# Mutations associated with neuropsychiatric conditions delineate functional brain connectivity dimensions contributing to autism and schizophrenia

Clara A. Moreau [1,2,14✉], Sebastian G. W. Urchs [2,3,14✉], Kumar Kuldeep [1], Pierre Orban[4,5], Catherine Schramm[1,6], Guillaume Dumas [1,7], Aurélie Labbe[8], Guillaume Huguet[1], Elise Douard [1], Pierre-Olivier Quirion[2,9], Amy Lin [10], Leila Kushan [10], Stephanie Grot[4,5], David Luck[1], Adrianna Mendrek[11], Stephane Potvin[5], Emmanuel Stip [5,12], Thomas Bourgeron[7], Alan C. Evans[3], Carrie E. Bearden [10,15], Pierre Bellec[2,15] & Sebastien Jacquemont [1,13,15✉]

16p11.2 and 22q11.2 Copy Number Variants (CNVs) confer high risk for Autism Spectrum Disorder (ASD), schizophrenia (SZ), and Attention-Deficit-Hyperactivity-Disorder (ADHD), but their impact on functional connectivity (FC) remains unclear. Here we report an analysis of resting-state FC using magnetic resonance imaging data from 101 CNV carriers, 755 individuals with idiopathic ASD, SZ, or ADHD and 1,072 controls. We characterize CNV FC-signatures and use them to identify dimensions contributing to complex idiopathic conditions. CNVs have large mirror effects on FC at the global and regional level. Thalamus, somato-motor, and posterior insula regions play a critical role in dysconnectivity shared across deletions, duplications, idiopathic ASD, SZ but not ADHD. Individuals with higher similarity to deletion FC-signatures exhibit worse cognitive and behavioral symptoms. Deletion similarities identified at the connectivity level could be related to the redundant associations observed genome-wide between gene expression spatial patterns and FC-signatures. Results may explain why many CNVs affect a similar range of neuropsychiatric symptoms.

[1] Sainte Justine Hospital Research Center, University of Montreal, 3175 Chemin de la Côte-Sainte-Catherine, Montreal, QC H3T 1C5, Canada. [2] Centre de Recherche de l'Institut Universitaire de Gériatrie de Montréal, 4565 Queen Mary Rd, Montreal, QC H3W 1W5, Canada. [3] Montreal Neurological Institute and Hospital, McGill University, 3801 Rue de l'Université, Montreal, QC H3A 2B4, Canada. [4] Centre de Recherche de l'Institut Universitaire en Santé Mentale de Montréal, 7401 Rue Hochelaga, Montreal, QC H1N 3M5, Canada. [5] Département de Psychiatrie et d'Addictologie, Université de Montréal, Pavillon Roger-Gaudry, C.P. 6128, succursale Centre-ville, Montreal, QC H3C 3J7, Canada. [6] Lady Davis Institute for Medical Research, Jewish General Hospital, 3755 Chemin de la Côte-Sainte-Catherine, Montreal, QC H3T 1E2, Canada. [7] Human Genetics and Cognitive Functions, Institut Pasteur, Université de Paris, UMR3571 CNRS Paris, France. [8] Département des Sciences de la Décision, HEC, 3000, chemin de la Côte-Sainte-Catherine, Montreal, QC H3T 2A7, Canada. [9] Canadian Center for Computational Genomics, McGill University and Genome Quebec Innovation Center 740, Dr. Penfield Avenue, H3A 0G1 Montreal, Canada. [10] Semel Institute for Neuroscience and Human Behavior and Department of Psychology, University of California, Los Angeles, Semel Institute/NPI, 760 Westwood Plaza, Los Angeles, CA 90024, USA. [11] Department of Psychology, Bishop's University, 2600 College Street, Sherbrooke, QC J1M IZ7, Canada. [12] United Arab Emirates University, College of Medicine and health Sciences, PO 17666, Al Ain, QC, UAE. [13] Department of Pediatrics, University of Montreal, 3175 Chemin de la Côte-Sainte-Catherine, Montreal, QC H3T 1C5, Canada. [14]These authors contributed equally: Clara A. Moreau, Sebastian G. W. Urchs. [15]These authors jointly supervised this work: Carrie E. Bearden, Pierre Bellec, Sebastien Jacquemont. ✉email: clara.moreau@umontreal.ca; sebastian.urchs@mail.mcgill.ca; sebastien.jacquemont@umontreal.ca

Copy number variants (CNVs) are deletions or duplications of DNA segments and represent an important source of genetic variation. An increase in rare CNV burden has been linked to a range of neurodevelopmental and psychiatric conditions[1,2]. Twelve recurrent CNVs have been individually associated with autism spectrum disorder (ASD)[3], eight with schizophrenia (SZ)[4], and eight with attention-deficit- hyper-activity disorder (ADHD)[5] but the mechanisms by which they lead to neuropsychiatric disorders remain unclear. Although they have large impacts on neurodevelopment, their effect alone does not lead to a psychiatric diagnosis. CNVs could, therefore, be leveraged to identify major dimensions contributing to complex idiopathic conditions.

CNVs at the proximal 16p11.2 and 22q11.2 genomic loci are among the most frequent large effect-size genomic variants and alter the dosage of 29 and 50 genes, respectively[6,7]. They confer high risk for ASD (10-fold increase for the 16p11.2 deletion and duplication)[3], SZ (>10-fold increase for the 22q11.2 deletion and 16p11.2 duplication)[4], and ADHD[8–12]. Gene dosage (deletions and duplications) affect the same neuroimaging measures in opposite directions (mirror effect). Structural alterations of the cingulate, insula, precuneus, and superior temporal gyrus overlap with those observed in meta-analytical maps of idiopathic psychiatric conditions including ASD and SZ[10,11].

Large effect-size mutations can shed light on pathways connecting genetic risk to brain endophenotypes, such as functional connectivity (FC). FC represents the intrinsic low-frequency synchronization between different neuroanatomical regions. It is measured by means of resting-state functional magnetic resonance imaging (rs-fMRI) which captures fluctuations of blood oxygenation as an indirect measure of neural activity across brain areas when no explicit task is performed[13,14]. Robust functional brain networks measured by rs-fMRI are also recapitulated by spatial patterns of gene expression in the adult brain[15,16].

Few studies have investigated the effect of 'neuropsychiatric' CNVs on FC. Dysconnectivity of thalamic-hippocampal circuitry[17] has been reported in 22q11.2 deletion carriers, with prominent under-connectivity of the default mode network (DMN), which was predictive of prodromal psychotic symptomatology[18,19]. Impaired connectivity of long-range connections within the DMN has also been reported by other studies[20]. A single 16p11.2 study has shown a decrease in connectivity of frontotemporal and -parietal connections in deletion carriers[21]. These initial studies have focussed on regions of interest but connectome-wide association studies (CWAS) analysing all connections without a priori hypotheses have not yet been performed in CNV carriers. Furthermore, their relation to idiopathic conditions has not been investigated.

Brain intermediate phenotypes of psychiatric conditions have mainly been studied by adopting a *top-down* approach, starting with a clinical diagnosis and moving to underlying neural substrates and further down to genetic factors[22]. Studies applying this analytical strategy in ASD have repeatedly shown patterns of widespread under-connectivity with the exception of over-connectivity in cortico-subcortical connections, particularly involving the thalamus[23,24]. SZ also exhibits a general under-connectivity profile, mainly involving the medial prefrontal cortex, the cingulate, and the temporal lobe[25], with over-connectivity of the thalamus[26]. These altered networks do not appear to be disorder-specific and have been reported across several disorders, including ASD, ADHD, and SZ[27]. These similarities seem to be distributed across several continuous dimensions[28] which may be related to shared genetic contribution across diagnoses, which is documented for common[29] and rare[30] variants, including the 16p11.2 and 22q11.2 CNVs.

We posit that seemingly distinct genetic variants and idiopathic disorders have overlapping patterns of dysconnectivity, which may help identify FC dimensions, providing insight into the complex connectivity architecture involved in psychiatric conditions.

We aimed to (1) characterize the FC-signatures of four high-risk neurodevelopmental CNVs, (2) explore whether FC-signatures of CNVs represent dimensions observed in idiopathic ASD, SZ, or ADHD, and (3) investigate the relationship between deletions at the FC and gene expression level.

To this end, we performed CWAS studies on 101 carriers of a 16p11.2 or 22q11.2 CNV, 122 of their respective controls, 755 individuals with idiopathic ASD, SZ, or ADHD and 950 of their respective controls.

## Results

**CNVs have effects on global and regional connectivity**. The 16p11.2 deletion showed a global increase in FC compared to controls with a mean shift = 0.29 z-scores ($p = 0.048$, permutation test, Fig. 1a, c; Supplementary Data 1.1). We observed 88 significantly altered connections (FDR, $q < 0.05$), and all but one were overconnected with beta values ranging from 0.76 to 1.34 z-scores. Overconnectivity predominantly involved the frontoparietal, somatomotor, ventral attention, and basal ganglia networks (Fig. 1c). Regions showing the strongest mean connectivity alterations included the caudate nucleus, putamen, lateral frontal pole, anterior middle frontal gyrus, and dorsal anterior cingulate cortex (Supplementary Data 1.8).

The 22q11.2 deletion was associated with a non-significant global decrease in connectivity Supplementary Data 1.3), with 68 negative connections surviving FDR correction (beta values ranging from −0.68 to −1.64 z-scores, Fig. 1b, d) when compared with control subjects. Underconnectivity predominantly involved the anterior and lateral DMN, and limbic network (Fig. 1d). The temporal pole, the ventral anterior insula and peri-insular sulcus, the amygdala-hippocampal complex, the dorsal anterior cingulate cortex, and perigenual anterior cingulate cortex showed the strongest changes in connectivity (see Supplementary Data 1.8).

For 16p11.2 duplication carriers, none of the individual connections survived FDR correction (Fig. 1a and Supplementary Data 1.2) relative to controls and none of the individual connections survived FDR correction. For the 22q11.2 duplications, only 16 connections survived FDR and these included overconnectivity in the posterior medial and lateral visual network, the cerebellum I-V, and the lateral fusiform gyrus (Fig. 1c and Supplementary Data 1.4 and 1.8).

Deletions and duplications at both loci showed a mirror effect at the global connectivity level. 16p11.2 deletions and duplications also showed mirror effects at the network level ($p = 0.006$, two-sided). This was not the case for 22q11.2 (Supplementary Notes).

A sensitivity analysis showed that results are unaffected by differences in age distribution between deletions and control groups as well as the number of remaining frames available after scrubbing (see Supplementary Notes).

**Effects of CNVs on FC are twice as large as those of SZ and ASD**. We performed three independent CWAS, comparing FC between patients with ASD, SZ, ADHD, and their respective controls. Idiopathic SZ showed overall underconnectivity affecting 835 connections, in line with previous reports[26,31] (Fig. 2a, c, Supplementary Notes and Supplementary Fig. 4, Supplementary Data 1.6 and 1.8). Over-connectivity was restricted to 24 connections (FDR, $q < 0.05$).

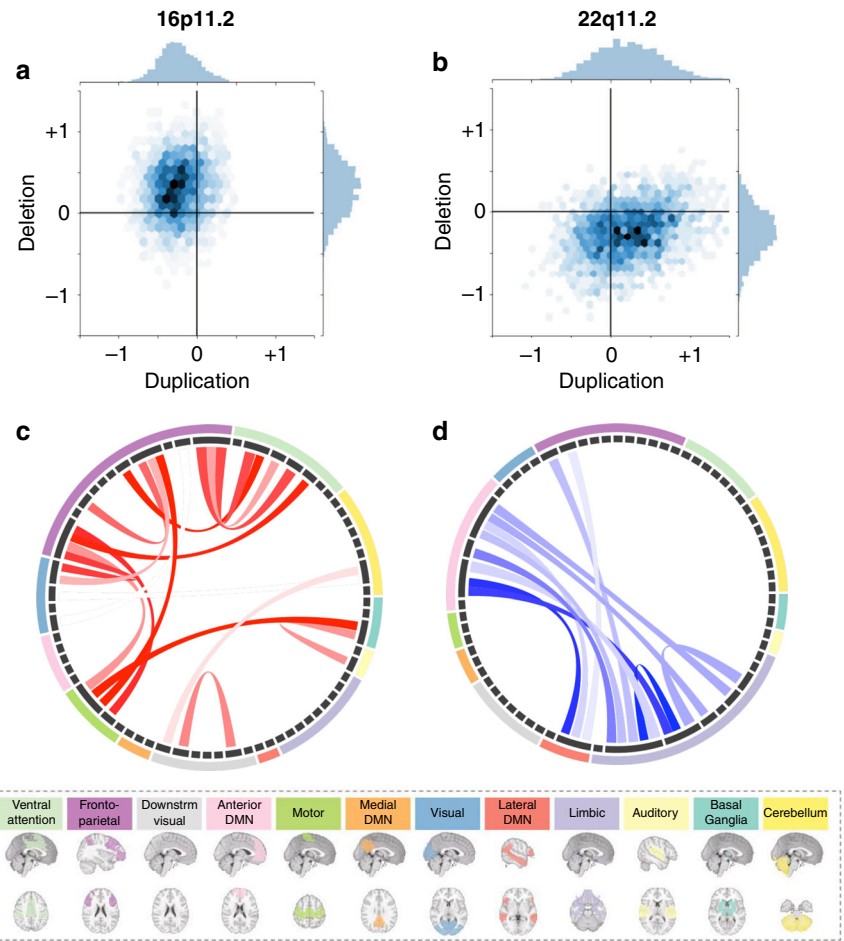

**Fig. 1 Connectome-wide effects of CNVs. a, b** Scatterplot (*hexagonal plot*), showing estimates (beta values) from connectome-wide association studies (CWAS) performed between 16p11.2 (**a**) and 22q11.2 (**b**) CNVs and their respective controls. In total, 2080 beta estimates were obtained from a linear model computed from z-scored connectomes based on the variance of the respective controls. The color hue represents the number of beta estimates in the hexagon bin. *Y*-axis: beta values associated with deletions (CWAS comparing deletions *vs* controls). *X*-axis: beta-values associated with duplications (CWAS comparing duplications *vs* controls). **c, d** Each chord diagram shows the top 20% of connections surviving FDR correction (*q* < 0.05) from the 16p11.2 deletion (**c**) and 22q11.2 deletion (**d**) CWAS. Each chord represents a significantly altered connection between two functional seed regions. All 64 seed regions are represented in the dark gray inner circle. The width of the seed region in the gray inner circle corresponds to the number of altered connections. Seed regions are grouped into 12 functional networks (outer ring, Supplementary Data 1.9). Networks are represented in 12 brains below the two diagrams. Red chords represent overconnectivity and blue chords underconnectivity.

Idiopathic ASD also showed overall underconnectivity (73 under and 2 overconnected survived FDR, *q* < 0.05, Fig. 2b, c, Supplementary Notes and Supplementary Figure 4, Supplementary Data 1.5 and 1.8).

For ADHD, none of the individual connections survived FDR correction (Supplementary Notes and Supplementary Data 1.7 and 1.8). Sensitivity analyses excluding females from the SZ and ADHD cohorts showed identical results (Supplementary Notes).

Among idiopathic conditions, the effect size of connectivity alteration was the highest in SZ (largest beta value = −0.56 std of the control group), followed by autism (largest beta value = −0.46), and ADHD (largest beta value = +0.26). Effect sizes observed for both deletions were approximately two-fold larger (beta values = +1.34 and −1.64 for 16p11.2 and 22q11.2 respectively) than those observed in idiopathic SZ, ASD, and ADHD (Fig. 2c). The largest effect size among the 16 connections surviving FDR for the 22q11.2 duplication was Cohen's *d* = 1.87.

**CNV-FC signatures show similarities with ASD and SZ.** We tested the spatial similarity between whole-brain FC-signatures across CNVs and idiopathic psychiatric conditions. To this mean

we computed the similarity (Pearson *R*) between group-level FC-signatures and the individual connectomes of either cases or controls from another group (Fig. 3). This was repeated 42 times between all CNVs and conditions and in both directions. Most of the significant whole-brain FC similarities were observed between individuals with either idiopathic ASD, SZ, and 4 CNVs (Fig. 2d). ADHD did not show any significant similarities with any other group.

**Thalamo-sensorimotor dysconnectivity across CNVs and psychiatric conditions.** We asked if whole-brain FC similarities between individuals with ASD, SZ, and CNVs may be driven by particular regions. We thus repeated the same similarity analysis presented above at the level of the FC signatures of each of the 64 seed regions. Individuals with SZ showed increased similarity with 28 out of the 64 regional FC-signatures of the 16p11.2 deletion than controls (FDR, *q* < 0.05). They also showed increased similarity with 18 region-level FC-signatures of the 22q11.2 deletion (Fig. 4, Supplementary Data 2.3 and 2.4, and Supplementary Notes). Deletion FC-signatures did not show any similarity with controls. We ranked the effect size of each seed

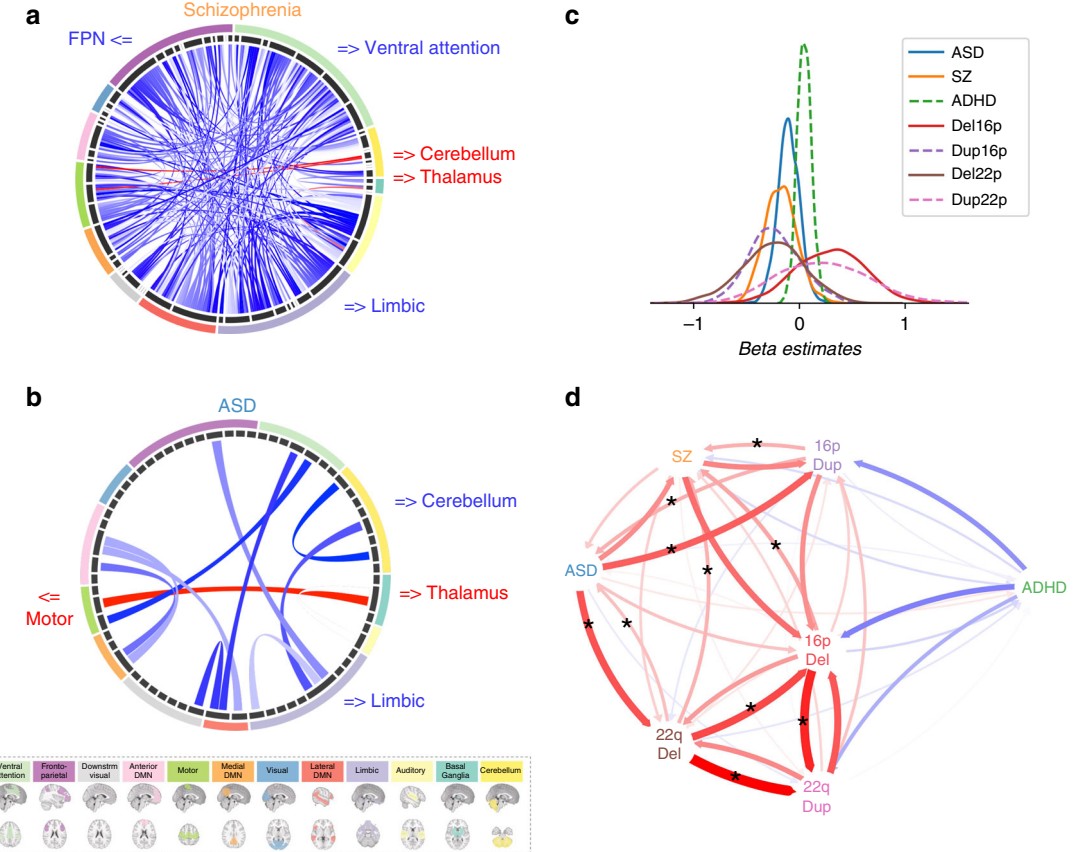

**Fig. 2 Connectome-wide similarity across ASD, SZ, and deletions. a, b** Each chord diagram shows the top 20% connections surviving FDR correction ($q <$ 0.05) from the SZ (**a**) and ASD (**b**) CWAS. Each chord represents a significantly altered connection between two functional seed regions. All 64 seed regions are represented in the dark gray inner circle. The width of the seed region in the gray inner circle corresponds to the number of altered connections. Seed regions are grouped into 12 functional networks (outer ring, Supplementary Data 1.9). The network colors correspond to the legend below. Red chords represent overconnectivity and blue chords underconnectivity. **c** Density plots represent the distribution of 2080 beta estimates for the CWAS (whole brain contrast of cases versus controls) for the SZ, ASD, ADHD, deletion and duplication groups. *X*-axis values = z-scores of Beta estimates, which were obtained from linear models computed using z-scored connectomes based on the variance of the respective controls. **d** The spatial similarity of whole-brain FC-signatures between CNVs and idiopathic psychiatric conditions. Arrows represent the correlation between group-level FC-signatures and the individual connectomes of either cases or controls from another group. The correlation was computed in both directions. The Red and blue arrows represent positive and negative correlations respectively. Arrow thickness represents the effect size of the Mann–Whitney test. Stars represent similarities (Mann–Whitney tests) surviving FDR. ASD autism spectrum disorder, SZ schizophrenia, ADHD attention-deficit hyperactivity disorder, FPN fronto-parietal network, 16pDel 16p11.2 deletion, 22qDel 22q11.2 deletion, 16pDup 16p11.2 duplication, 22qDup 22q11.2 duplication.

region and compared them for both deletions. The seed regions with the highest similarity between SZ and 16p11.2 were also those with the highest similarity between SZ and 22q11.2 (adjusted $R^2 = 0.37$, $p = 6e{-}08$) (see Supplementary Notes). Sensitivity analysis showed that the same regions are driving similarities with SZ and ASD irrespective of psychiatric diagnoses in 22q11.2 deletion carriers (see Supplementary Notes).

Individuals with autism showed greater similarity with six regional FC-signatures of the 16p11.2 deletion compared to controls (FDR, $q < 0.05$). They also showed greater similarity with six region-level FC-signatures of the 22q11.2 deletion (Fig. 4, Supplementary Data S2.1 and 2.2, and Supplementary Notes). Deletion FC-signatures did not show significant similarities with controls for any of the 64 seed regions. Of note, individuals with SZ and ASD showed higher similarity with the thalamus FC-signatures of both deletions (Fig. 4). None of the similarities correlated with motion or sex. Regions driving similarities between psychiatric conditions and deletion FC signatures were also those with the highest number of connections altered by each deletion individually. Eight and six out of the top 10 regions altered by 22q11.2 and 16p11.2 respectively were

driving similarities with psychiatric conditions (Supplementary Data 1.8).

Despite lower power, we investigated similarities with duplication FC-signatures. The number of significant regional similarities was smaller. Out of the 28 regions showing a similarity between idiopathic conditions and duplications, 17 regions also showed similarities with deletions (See Supplementary Notes). Individuals with ADHD did not show higher similarities with the regional FC-signatures of any CNVs except for 2 regional FC-signatures of the 16p11.2 duplication (Supplementary Data 2.5 and 2.6).

**Similarity with CNV-FC signatures is associated to symptom severity.** We investigated whether regional FC similarities with deletions described above are associated with symptom severity among individuals in idiopathic psychiatric cohorts. Symptom severity was assessed using the Autism Diagnostic Observation Schedule (ADOS[32]), in ASD, Positive and Negative Syndrome Scale (PANSS[33]) in SZ, and Full-Scale Intelligence Quotient (FSIQ) in ASD. The 10 seed regions with significant FC similarity

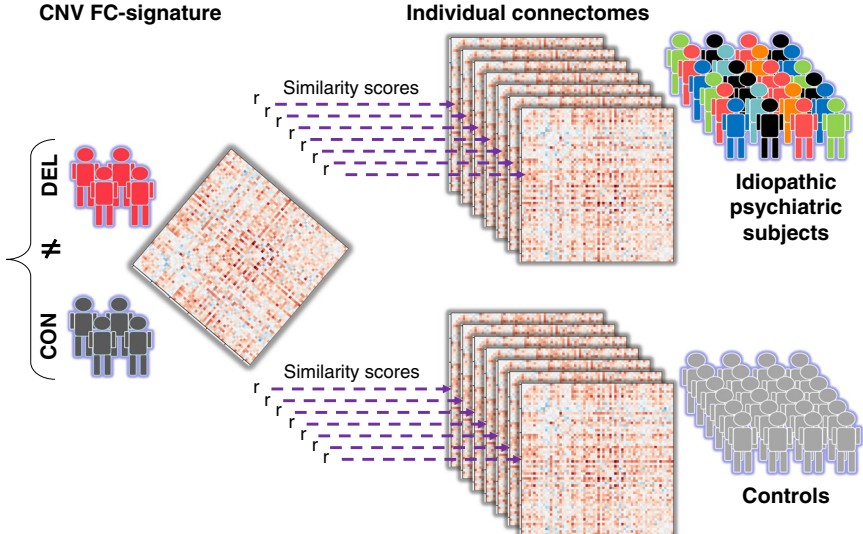

**Fig. 3 Testing similarities across CNVs and idiopathic conditions.** Similarities of FC-signatures across this study were characterized by correlating (Pearson's *r*) a group-level FC-signature with individual connectomes from either cases and controls. The *r* values obtained for all cases and all controls were compared using a Mann–Whitney test. Here, the group level connectome is represented by a matrix of 2080 beta values, on the left side. It is obtained by contrasting deletion cases (red) and controls (dark gray). The beta map is correlated to seven individual connectomes of psychiatric cases and seven connectomes of controls. The different colors used for psychiatric cases represent phenotypic heterogeneity.

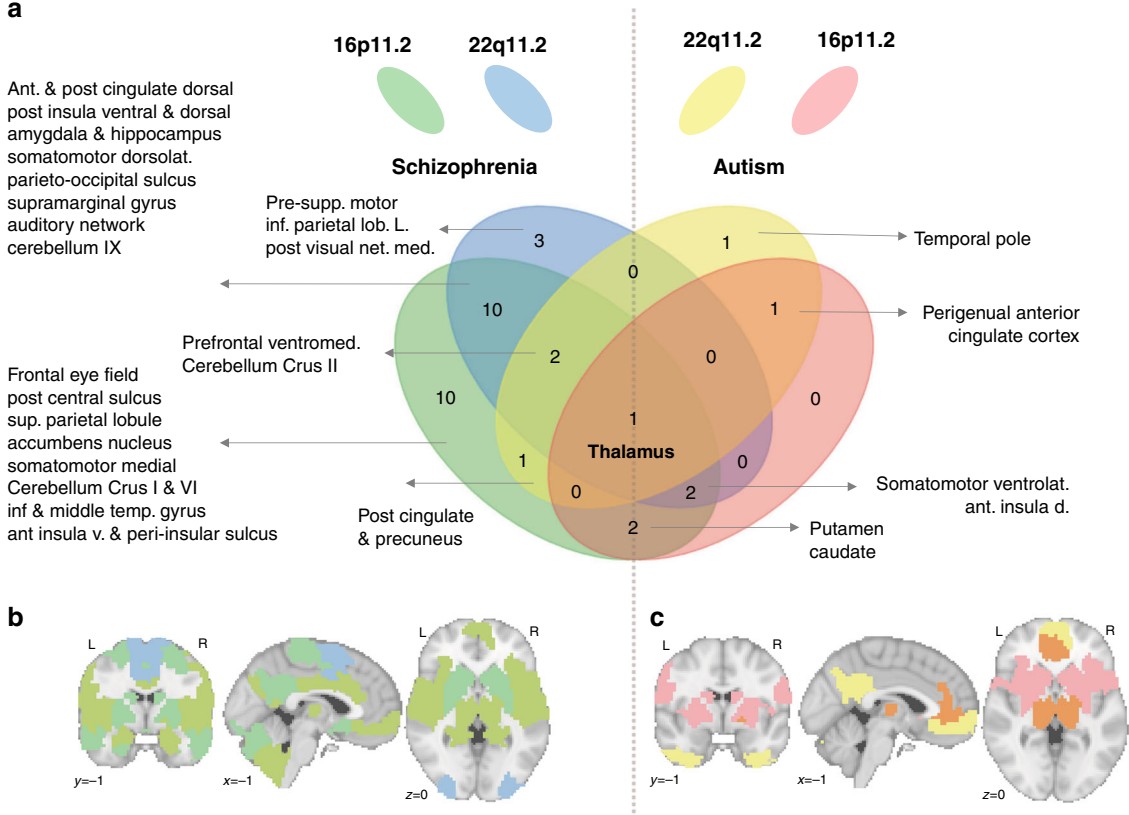

**Fig. 4 Regional FC similarity between psychiatric diagnoses and deletions.** The FC-signatures of both deletions are decomposed into 64 seed-regions. Deletion FC-signatures are correlahe individual connectivity profile of subjects with a psychiatric diagnosis and their respective control subjects. Of note, the correlation is equivalent to the mean centering of all region-based FC-signatures. Significantly higher similarities of patients with either ASD and SZ were present in 33 seeds regions (FDR) and are presented on the right and the left side of the Venn diagram, respectively (**a**), and also in the corresponding left (**b**) and right (**c**) brain maps. At the intersection of all ellipses, the thalamus FC-signatures of both deletions showed increased similarity with individuals who have a diagnosis of ASD or SZ compared to their respective controls. 16pDel 16p11.2 deletion, 22qDel 22q11.2 deletion, Ant anterior, Post posterior, dorsolat dorsolateral, Inf inferior, L. left, v. ventral, net. network, med medial, Supp supplementary, lob lobule (Full-name labels are provided in Supplementary Data 1.9).

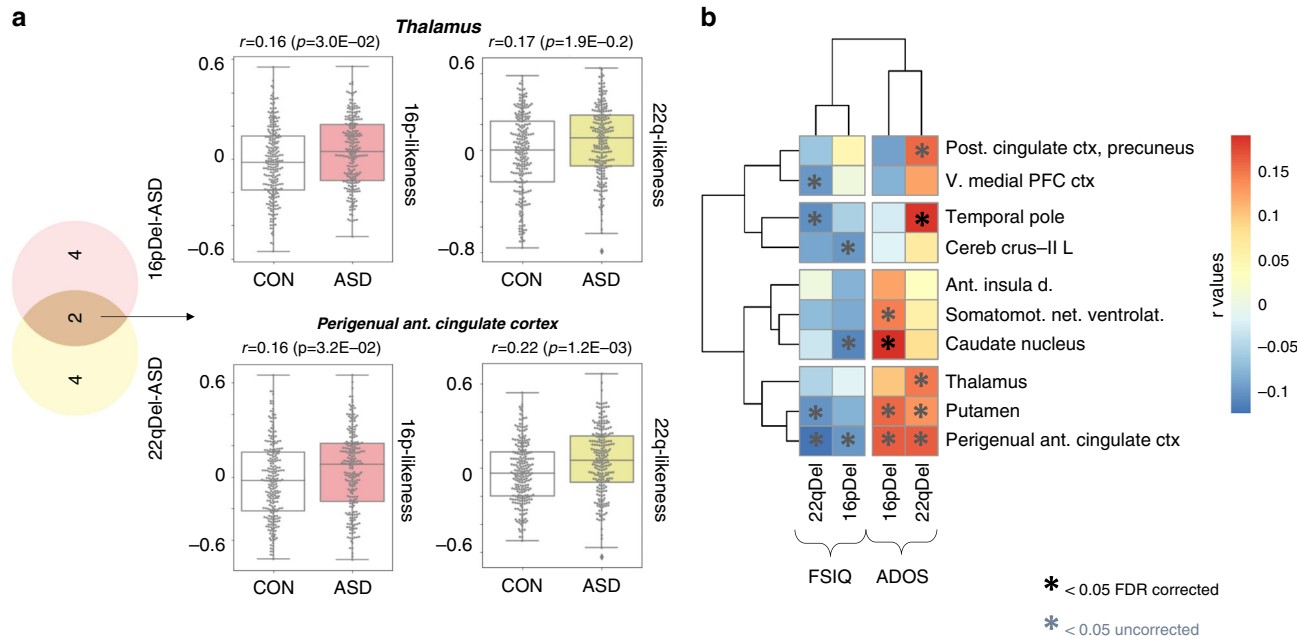

**Fig. 5 Relationship between the deletion FC-signatures and behavior. a** Boxplots represent the connectivity similarity for two seed regions (thalamus and perigenual anterior cingulate cortex). Boxplots display the 25th, 50th, and 75th percentile of the underlying data. Boxplot whiskers encompass data points within 1.5 times the interquartile range from the median line and more extreme data points are labeled with outlier fliers. Each data point represents one individual: *r*-value of the Pearson correlation between the deletion FC-signatures and the FC-profile of an individual with ASD (*n* = 225 in the colored boxplots) or a control subject (*n* = 234 in the non-colored boxplots). For the two seed regions, individuals with ASD show significantly higher similarity (FDR, *q* < 0.05) with the 16p11.2 (pink) and 22q11.2 (yellow) deletion FC-signatures than the individual controls. All seed regions showing significantly higher similarity with ASD are represented in the Venn diagram on the left. **b** We investigated the relationship with cognitive scores and found that stronger individual similarity (Pearson's *r*) with the deletion FC-signature was associated with more severe symptoms measured by FSIQ and ADOS. Heatmaps show the level of correlation (FDR, *q* < 0.05, two-sided) between behavior scores and the similarity with deletion FC-signatures. 16pDel 16p11.2 deletion, 22qDel 22q11.2 deletion, FSIQ full-scale intelligence quotient, ADOS autism diagnostic observation schedule, ant. anterior, post posterior, v. ventral, PFC prefrontal cortex, cereb cerebellum, d. dorsal, L left, ctx cortex, net. network. Full-name labels are provided in Supplementary Data 1.9.

between ASD and either deletion were those showing the strongest association with the ADOS symptom-severity score (two regions passed FDR correction *q* < 0.05: the caudate nucleus and temporal pole) and FSIQ (Fig. 5 and Supplementary Data 3.1–3.4). Among the seed regions contributing to the similarity between SZ individuals and deletions, none were significantly associated with PANSS measures after FDR correction. FSIQ data were not available in the SZ cohorts.

**16p11.2 and 22q11.2 deletions show regional FC similarities.** Although the two deletions showed opposing effects on global connectivity (Fig. 2c), their connectomes were positively correlated (Fig. 2d). We, therefore, sought to identify the main regions that contributed to this connectome-wide correlation.

Using the same approach as above (Fig. 3), we correlated the 22q11.2 deletion group-level FC-signature with individual connectomes of 16p11.2 deletion carriers and their respective controls. The 22q11.2 deletion FC-signature showed significant similarities with 16p11.2 deletion carriers for 12 regions (FDR, *q* < 0.05), mainly involving the frontoparietal, ventral attentional, and somatomotor networks. The reverse test showed significant similarity of the 16p11.2 deletion FC-signature with 22q11.2 deletion carriers in 10 regions within the anterior and lateral DMN, frontoparietal and basal ganglia networks. Four seed regions were observed in both tests (Fig. 6a). We reasoned that the FC-similarity between CNVs may be informed by the spatial patterns of gene expression within both genomic intervals.

**CNV FC-signatures associated with gene expression patterns.** We performed partial least squares regression (PLSR) to

investigate the association between FC-signatures of each deletion and the expression patterns of 37 and 24 genes encompassed in the 22q11.2 and 16p11.2 genomic loci respectively. The two components required to reach a significant association explained 24.2% of the variance of the 16p11.2 deletion FC profile (*p* = 0.041, 5000 random FC profiles). For the 22q11.2 deletion, either one or two components were significant (*p* < 0.0002, 5000 random FC profiles). The two components explained 43.2% of the variance of the 22q11.2 deletion FC signature.

Similar PLSR analyses performed for each of the 64 regions showed that 18 and 32 regional FC-signatures were significantly associated (5000 random FC-signatures, FDR 64 regions) with spatial patterns of gene expression at the 16p11.2 and 22q11.2 loci respectively (Fig. 6b, c). However, this relationship was not specific because 22q11.2 genes were also associated with *n* = 20 regions of the 16p11.2 FC signature. Conversely, the 16p11.2 genes were associated with *n* = 19 regions of the 22q11.2 FC signature (Fig. 6c, Supplementary Data S4.2). Overall, this relationship was repeatedly observed across the genome: a large proportion of gene sets (randomly sampled genome-wide, *n* = 37 and 24) showed a similar level of association with either global or regional FC-signature of both deletions especially for the 22q11.2.

To further investigate the low specificity of the connectivity/gene expression relationship, we tested the individual correlation (Pearson) of all 15663 genes with available expression data with deletion FC signatures (Fig. 6d, e, Supplementary Data 4.3–4.4). Correlations with the 16p11.2 and 22q11.2 FC signatures were observed for 421 and 3883 genes respectively (5000 random FC-signatures). After genome-wide FDR correction, the expression of 1834 genes remained spatially correlated with 22q11.2 and none

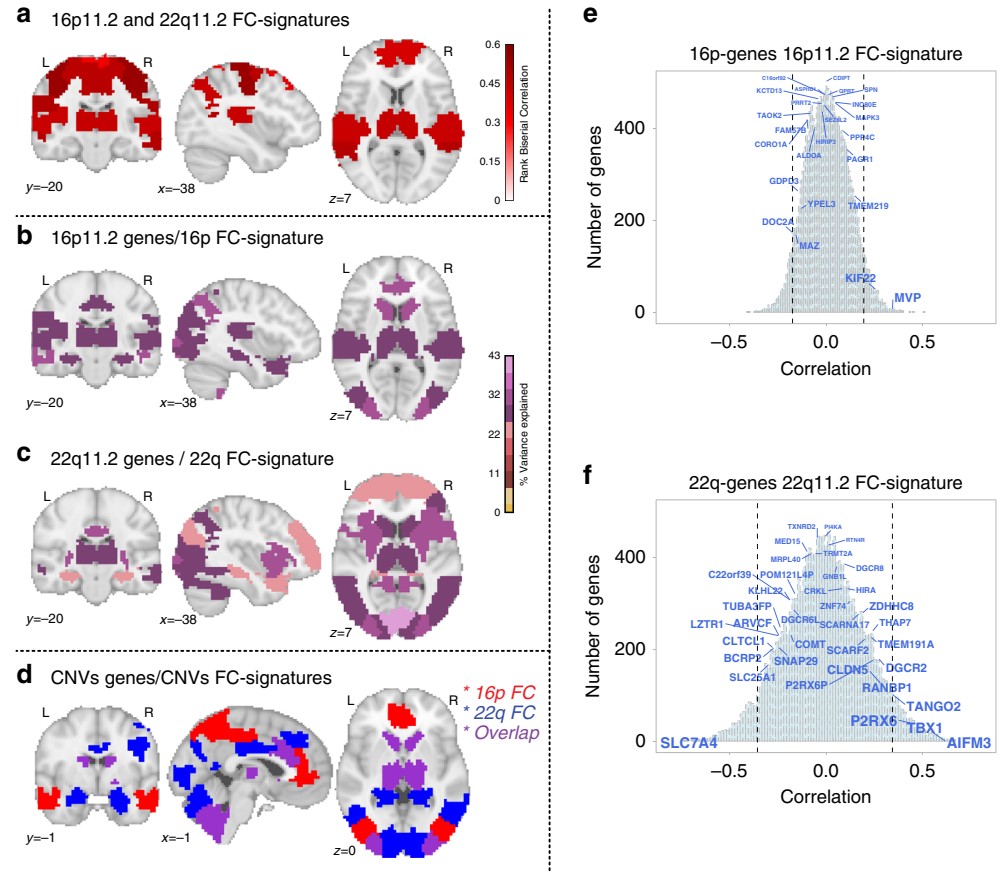

**Fig. 6 FC similarities between CNVs and relationship with gene expression. a** FC similarities between both deletions at the regional level. The values in the brain map represent the level of the FC similarity between deletions (rank biserial correlation, Mann–Whitney test). The values are thresholded (FDR, 64 regions): 18 out of 64 regions are similar between deletions. **b** Relationship between spatial patterns of gene expression within the 16p11.2 locus and regional FC signatures of the 16p11.2 deletion. A partial least square regression (PLSR) was conducted for each of the 64 regions. Maps are thresholded (FDR corrected for 64 regions) and color code represents the percentage of variance explained by gene expression using two components in the PLSR. **c** The same analysis was conducted for 22q11.2 genes and the 22q11.2 deletion FC signature. **b, c** Eleven regions overlapped across PLSR maps: thalamus, caudate, anterior insula and posterior insula sulcus, amygdala and hippocampus, cerebellum 9 and right crus-2, dorsal anterior cingulate, left inferior parietal lobule, medial posterior visual network, lateral posterior visual network, and dorsal visual network. Three (over the 18) regions identified in the between deletion similarity (**a**) analysis are also present in the gene expression/FC-signature association maps (**b, c**): Thalamus, dorsal anterior cingulate, and left inferior parietal lobule. **d** Low specificity for the relationship between spatial patterns of gene expression and regional FC deletion signatures. In red: the 16p11.2 regional FC associated with the expression patterns of both the 16p11.2 and the 22q11.2 genes. In blue, the 22q11.2 regional FC is associated with the expression patterns of genes in both genomic loci. In purple, seven regions were found with both deletion FC-signatures, and the expression patterns of genes encompassed in both genomic loci. **e, f** Expression patterns of genes within and outside CNVs correlate with FC-signatures of 16p11.2 and 22q11.2 deletions. The light blue histogram represents the distribution of correlations for 15663 genes with available gene expression data from the AHBA. X-axis values: Pearson coefficients. Y-axis values: number of genes. Genes within the CNVs have font-size scaled based on p values. Dotted lines represent the 5th and 95th percentiles of the correlation distribution genome-wide.

with the 16p11.2 deletion FC signature. The median correlation values for the $n = 24$ 16p11.2 genes and the 16p11.2 FC signature was not higher than the median correlation of 10000 randomly sampled gene sets of the same size ($n = 24$, $p = 0.31$). The same was true for 22q11.2 genes ($n = 37$, $p = 0.36$). However, genes with correlations ranking higher than the genome-wide 98th percentile were over-represented in both deletions: *MVP* and *KIF22* showed correlations ($r_{MVP} = 0.33$, $r_{KIF22} = 0.26$, Supplementary Data S4.4) ranking at the 99.76th and 98.76th percentile genome-wide ($p = 0.03$; null:10,000 random gene sets). For 22q11.2, *AIFM3, TBX1,* and *P2RX6* showed correlations in the 99.5, 98.84, 98.33th percentile ($p = 0.04$).

## Discussion
This proof of concept study provides the first connectome-wide characterization of four CNVs that confer high risk for

psychiatric disorders. Deletions and duplications at the 16p11.2 and, to a lesser extent, the 22q11.2 locus were associated with mirror effects at the connectome-wide level. Overconnectivity in the 16p11.2 deletion predominantly involved the ventral attention, motor, and frontoparietal networks. Underconnectivity in the 22q11.2 deletion involved the anterior and lateral DMN and the limbic network. Regional FC-signatures of deletions and duplications, in particular, those implicating the thalamus, somatomotor, posterior insula and cingulate showed significant similarities with the complex architecture of idiopathic ASD, SZ but not ADHD. Seemingly distinct, rare neuropsychiatric mutations may converge on dimensions representing mechanistic building blocks shared across idiopathic conditions. The spatial expression patterns of genes encompassed in both genomic loci were associated with FC-signatures of the corresponding deletion but many genes outside these 2 loci also show similar levels of

association. This redundancy may represent a factor underlying shared FC signatures between both deletions.

22q11.2 and 16p11.2 CNVs showed large effect sizes on FC that are similar to those previously reported for structural neuroimaging measures, cognition, and behavior[8,10,11]. In sharp contrast, there is significant discordance between the severe clinical manifestations observed in idiopathic ASD and SZ, and the small effect size observed in case-control studies at the FC level. Previous structural neuroimaging studies of the same idiopathic psychiatric conditions have also reported small effect sizes[34,35]. This discordance may be due to the heterogeneity of these idiopathic conditions and hints at the presence of subgroups or latent dimensions associated with larger effect sizes[28].

The FC-signatures of both deletions (and to a lesser extent those of both duplications) showed similarities with autism and SZ, but not ADHD. Regions contributing to these similarities were also those with the highest number of connections altered by each deletion individually. The FC-signature of the same seed regions also showed the highest association with ASD severity scores and general intelligence in the idiopathic autism sample.

Among the regions, results highlighted overconnectivity between the thalamus and sensory-motor, auditory and visual networks as a common alteration across CNVs and individuals with idiopathic autism or SZ who do not carry CNVs. This is in line with recent rs-fMRI studies performed across psychiatric illnesses[28]. Perceptual dysfunctions are core features of SZ and ASD[36]. Those include auditory and visuals hallucinations in SZ[37,38], impairments in gestalt visual perception and discrimination of visual motion[39], disturbances in auditory and tactile discrimination in Autism. Impairments in phonology[40], as well as visual and auditory deficits, have also been demonstrated in 16p11.2 and 22q11.2 deletion carriers[41–43]. A general thalamo-sensory disturbance may, therefore, be central across genomic mutations and psychiatric diagnoses. Further studies are required to investigate genome-wide, the genetic determinants of thalamo-sensory disturbance. Because it appears ubiquitous across conditions, the genetic basis is likely to be very broad. FC similarities between idiopathic psychiatric disorders, deletions, and duplications are also in line with an emerging body of literature that points to common neurobiological substrates for mental illness[44]. Evidence includes the genetic correlation between psychiatric disorders[29,45] and pleiotropic effects of CNVs associated with several conditions[2,46].

Recent work has shown that many genes share similar spatial patterns of expression[15] organized along broad spatial gradients in the brain that are closely related to FC networks[47,48]. In line with these observations, we show that FC-signatures of deletions are associated with expression patterns of genes within as well as outside the genomic loci of interest. The FC profile of the thalamus, dorsal anterior cingulate, and left inferior parietal lobule were associated with expression patterns of genes at both loci and may explain, in part, the FC similarities between both deletions.

FC studies using a top-down case-control approach (eg. autism versus control) have characterized large-scale brain network changes associated with diseases, but this framework is unable to describe the directionality of this relationship[49]. FC-changes may not necessarily represent an intermediate brain phenotype but rather a secondary impact of psychiatric illnesses. Our strategy integrating top-down and bottom-up approaches show that individuals with idiopathic ASD or SZ as well as CNV carriers who do not meet diagnostic criteria for these conditions share regional FC alterations. This suggests that the risk conferred by genetic variants and the associated FC-patterns represent important dimensions that are necessary but insufficient to cause disease. Additional factors and associated FC-patterns are required (incomplete penetrance[50]). Bottom-up approaches

studying rare variants have almost exclusively been performed individually. Our results suggest, however, that they likely converge on overlapping intermediate brain phenotypes, consistent with a recent study showing overlapping effects on subcortical structures across 12 different CNVs conferring risk to SZ[51].

**Limitations**. Reproducibility of rs-fMRI in psychiatry has been challenging. However, when studies using similar analytical strategies are compared, there are consistent results. In SZ and ASD, global decrease in FC has been reported by most studies except for those adjusting for global signal[24,52]. Increased thalamocortical connectivity is also repeatedly reported in both conditions[24,26,31]. These previous findings are consistent with our results (see Supplementary Note). The 22q11.2 deletion FC signature is consistent with previous works on 22q11.2 FC alterations that showed (1) underconnectivity of the DMN[53,54], (2) thalamocortical overconnectivity and underconnectivity involving the hippocampus[17]. The only rs-fMRI study previously published for the 16p11.2 deletion focused on the dmPFC[21]. Using the same approach and regressing global signal, we also found underconnectivity of the dmPFC with the same set of regions. This highlights the fact that many seemingly discrepant results can be reconciled once methodologies are aligned.

There is no available genetic data for any of the three idiopathic cohorts. However, the frequency of 16p11.2 and 22q11.2 CNVs in ASD or SZ is <1%[3,4]. This suggests that the observed FC similarities between CNVs and ASD or SZ are driven by other factors.

Expression data were derived from 6 adult brains of the AHBA. Gene expression studies reported a later window of susceptibility in SZ compared to ASD[55]. The 22q11.2 deletion may, therefore, be in part driven by later onset mechanisms, which could explain why its FC-signature correlates to the adult spatial expression patterns of so many genes. On the other hand, the 16p11.2 deletion which is preferentially associated with autism may be linked to earlier developmental mechanisms which may explain why its FC-signature is much less correlated to patterns of adult brain expression. further analyses are required to understand the potential relationship between regional connectivity and temporal gene expression.

The results on duplications should be interpreted with caution due to our limited power to detect changes in connectivity. The limited phenotypic data in the SZ group did not allow to investigate the relationship between deletion FC-signatures and cognitive traits in this sample. Lack of similarity observed for ADHD is line with the small association between 16p11.2 CNVs and ADHD but is discordant with the association reported for 22q11.2[5]. ADHD has a smaller effect size than SZ and ASD, which may have limited our analysis[56]. Several confounding factors may have influenced some of the results. Those include sex bias, which is present across all three psychiatric cohorts, age differences in the 16p11.2 deletion group, diagnosis of ASD and ADHD in 22q11.2 deletion carriers, and medication status in the idiopathic ASD and SZ groups. However, carefully conducted sensitivity analyses, investigating all of these confounders did not change any of the results.

Deletion and duplication at several genomic loci result in mirror effects across many human traits[57–60], including brain connectivity. Haploinsufficiency may define FC dimensions that represent building blocks contributing to idiopathic psychiatric conditions. Thalamo-sensory disturbance may represent one dimension central across genomic mutations and psychiatric diagnoses. The redundant associations observed, genome-wide, between gene expression and connectivity may explain similarities across genomic variants and idiopathic conditions and the

**Table 1 Cohort characteristics.**

| Dataset | Status | n | Age | FSIQ | Sex | FD[a] | ASD[a] | ADHD | SZ |
|---|---|---|---|---|---|---|---|---|---|
| SVIP 16p11.2 Cohort (2 sites) | Del carriers | 20 | 12.7 (6.8) | 92.5 (16.1) | 12 M | 0.18 (0.03) | 4 | 6 | 0 |
| | Controls | 79 | 26.7 (14.7) | 103.6 (14.8) | 46 M | 0.17 (0.04) | 0 | 0 | 0 |
| | Dup carriers | 23 | 28.2 (12.8) | 94.1 (15.1) | 12 M | 0.19 (0.05) | 1 | 1 | 0 |
| UCLA 22q11.2 Cohort (1 site) | Del Carriers | 46 | 16.8 (6.1) | 77.2 (13.8) | 20 M | 0.17 (0.06) | 23 | 19 | 3 |
| | Controls | 43 | 13.0 (4.6) | 111.9 (17.6) | 22 M | 0.14 (0.04) | 0 | 2 | 0 |
| | Dup carriers | 12 | 16.74 (13) | 95.7 (19) | 7 M | 0.17 (0.1) | 3 | 4 | 0 |
| ASD Cohort (ABIDE 10 sites) | Cases | 225 | 15.9 (6.5) | 103.7 (17.4) | 225 M | 0.18 (0.05) | 225 | – | – |
| | Controls | 234 | 15.7 (6.1) | 110.63 (12.1) | 234 M | 0.17 (0.04) | 0 | – | – |
| Schizophrenia Cohort (10 sites) | Cases | 241 | 33.62 (9.2) | – | 179 M | 0.16 (0.06) | – | – | 241 |
| | Controls | 242 | 32.3 (9.6) | – | 181 M | 0.14 (0.05) | – | – | 0 |
| ADHD Cohort (7 sites) | Cases | 289 | 11.5 (2.8) | 106.8 (13.7) | 227 M | 0.15 (0.04) | – | 289 | – |
| | Controls | 474 | 12.2 (3.3) | 114.2 (13.1) | 250 M | 0.14 (0.04) | – | 0 | – |

Quantitative variables are expressed as the mean ± standard deviation.
Description of the cohorts after filtering for quality criteria. *SVIP* Simons Variation in Individuals Project, *UCLA* University of California, Los Angeles, *ASD* autism spectrum disorder, *ABIDE* Autism Brain Imaging Data Exchange, *SZ* schizophrenia, *ADHD* attention-deficit/hyperactivity disorder, *Del* deletion, *Dup* duplication; Age (in years), *FSIQ* Full-Scale Intelligence Quotient, *M* male, *FD* framewise displacement (in mm).
[a]More information regarding the remaining number of time frames for each group, and the percentage of motion censoring, is provided in Supplementary Information. Sensitivity analyses investigating sex bias in the three idiopathic cohorts are presented in Supplementary Notes. Sensitivity analysis investigating medication effect in ASD cohort is presented in Supplementary Notes. Sensitivity analyses also showed that the FC-signature of 22q11.2 deletions is not influenced by a diagnosis of ASD or ADHD (Supplementary Fig. 1). Columns ASD, SZ, and ADHD represent the number of subjects with those diagnoses. One subject may have several diagnoses. For example, 9 subjects with ASD have also an ADHD diagnosis.

fact that many CNVs affect a similar range of neuropsychiatric symptoms. It is, therefore, becoming increasingly difficult to justify the study of psychiatric conditions or rare genetic variants in isolation. Large-scale studies simultaneously integrating a top-down approach across diagnostic boundaries, and a bottom-up investigation across a broad set of genomic variants are required to improve our understanding of specific and common psychiatric outcomes associated with genetic variants and FC signatures.

## Methods

**Samples**. We performed a series of CWAS using individuals from five data sets (Table 1 and Supplementary Information).

(1–2) Two genetic-first cohort (recruitment based on the presence of a genetic variant, regardless of any DSM diagnosis):

16p11.2 deletion and duplication carriers (29.6–30.1 MB; Hg19), and extrafamilial controls from the Simons Variation in Individuals Project (VIP) consortium[61].

22q11.2 deletion and duplication carriers (18.6–21.5 MB; Hg19) and extrafamilial controls from the University of California, Los Angeles.

(3) Individuals diagnosed with ASD and their respective controls from the ABIDE1 multicenter dataset[23].

(4) Individuals diagnosed with SZ (either DSM-IV or DSM-5) and their respective controls. We aggregated fMRI data from 10 distinct studies.

(5) Individuals diagnosed with ADHD (DSM-IV) and their respective controls from the ADHD-200 dataset[62,63].

Imaging data were acquired with site-specific MRI sequences. Each cohort used in this study was approved by the research ethics review boards of the respective institutions. Signed informed consent was obtained from all participants or their legal guardian before participation. Secondary analyses of the listed datasets for the purpose of this project were approved by the research ethics review board at Sainte Justine Hospital. After data preprocessing and quality control, we included a total of 1,928 individuals (Table 1).

**Preprocessing and quality control procedures**. All datasets were preprocessed using the same parameters with the same Neuroimaging Analysis Kit version 0.12.4, an Octave-based open-source processing and analysis pipeline[64]. Pre-processed data were visually controlled for quality of the co-registration, head motion, and related artefacts by one rater (Supplementary Informations).

**Computing connectomes**. We segmented the brain into 64 functional seed regions defined by the multi-resolution MIST brain parcellation[65]. FC was computed as the temporal pairwise Pearson's correlation between the average time series of the 64 seed regions, after the Fisher transformation. The connectome of each individual encompassed 2,080 connectivity values: $(63 \times 64)/2 = 2016$ region-to-region connectivity + 64 within seed region connectivity. We chose the 64 parcel atlas of the multi-resolution MIST parcellation as it falls within the range of network resolution previously identified to be maximally sensitive to FC alterations in neurodevelopmental disorders such as ASD[66].

Statistical analyses were performed in Python using the scikit-learn library[67]. Analyses were visualized in Python and R. Code for all analyses and visualizations is being made available online through the GitHub platform https://github.com/surchs/Neuropsychiatric_CNV_code_supplement.

**Connectome-wide association studies**. We performed seven CWAS, comparing FC between cases and controls for four CNVs (16p11.2 and 22q11.2, either deletion or duplication) and three idiopathic psychiatric cohorts (ASD, SZ, and ADHD). Note that controls were not pooled across cohorts. Within each cohort, FC was standardized (z-scored) based on the variance of the respective control group. CWAS was conducted by linear regression at the connectome level, in which z-scored FC was the dependent variable and clinical status the explanatory variable. Models were adjusted for sex, site, head motion, and age. We determined whether a connection was significantly altered by the clinical status effect by testing whether the β value (regression coefficient associated with the clinical status variable) was significantly different from 0 using a two-tailed *t*-test. This regression test was applied independently for each of the 2080 functional connections. We corrected for the number of tests (2080) using the Benjamini-Hochberg correction for FDR at a threshold of $q < 0.05$[68], following the recommendations of Bellec et al.[69].

We defined the global FC shift as the average of the β values across all 2080 connections and tested for significance using a permutation test. We performed 5000 random CWAS by contrasting CNV carriers and controls after shuffling the genetic status labels. For example, we randomly permuted the clinical status of 16p11.2 deletion carriers and their respective controls in the 16p11.2 deletion *vs* control CWAS. We then estimated the *p*-value by calculating the frequency of random global FC shifts that were greater than the original observation[70].

**Gene dosage mirror effects on FC**. We tested whether networks are affected by gene dosage in a mirror fashion by computing the product of the β values obtained in each genetic group contrasts: "Deletions *vs* Controls" and "Duplications *vs* Controls" (separately for 16p11.2 and 22q11.2). Negative values indicate mirror effects of deletions and duplications on FC. Positive values indicate effects in the same direction for deletions and duplications. The obtained products of the β values were grouped into 12 canonical functional networks using information from the multi-resolution brain parcellation (Supplementary Data 1.9).

**Whole-brain FC-similarity between idiopathic psychiatric conditions and CNVs**. We tested the similarity between dysconnectivity measured across idiopathic psychiatric conditions and CNV. This similarity was tested by correlating individual whole-brain connectomes of cases and controls of one group to the whole-brain FC-signature (group level) of another group (Fig. 3). The group-level FC-signature was defined as the 2080 β values obtained from the contrast of cases vs. controls. This was repeated 21 times between all CNVs and conditions and in both directions ($n = 42$ similarity tests).

Individual connectomes of cases and their respective controls were used after independently adjusting for sex, site, head motion, age, and average group connectivity for each of the datasets.

Similarity scores were derived by computing Pearson's correlations between the whole-brain connectomes. We asked whether cases compared to their respective controls had significantly higher (or lower) similarity to whole-brain FC-signature of another group using a Mann–Whitney $U$ test. We reported significant group differences after FDR correction accounting for the 42 tests ($q < 0.05$).

**Similarity of regional FC-signatures between idiopathic conditions and CNVs.** The same approach described above was performed at the regional level. Each of the 1705 connectomes of individuals with idiopathic psychiatric conditions and their respective controls was independently adjusted for sex, site, head motion, age, and average group connectivity for each dataset. We calculated a similarity score between these individual connectomes and the FC-signatures of the 16p11.2 and 22q11.2 deletions and duplications. The FC-signatures were broken down into 64 region-level FC-signatures and similarity scores were derived by computing Pearson's correlations between the 64 β values associated with a particular region. For each region, we tested whether individuals with a psychiatric diagnosis had significantly higher (or lower) similarity to 16p11.2 or 22q11.2 deletion FC-signatures than their respective controls using a Mann–Whitney U test. We reported significant group differences after FDR correction ($q < 0.05$) for the number of regions (64).

We investigated the relationship between symptom severity and similarity with deletions. The similarity of individuals with deletion FC-signatures was correlated (Pearson's r) with cognitive and behavioral measures. Those included the ADOS and FSIQ in the autism sample and the PANSS in the SZ sample. The p-values associated with these correlations were corrected for multiple comparisons (FDR, $q < 0.05$).

**Similarity of 16p11.2 and 22q11.2 deletions signatures.** We correlated the 22q11.2 group-level deletion-FC-signature with individual connectomes of 16p11.2 deletion carriers and their respective controls. We correlated as well the 16p11.2 group-level deletion-FC-signature with individual connectomes of 22q11.2 deletion carriers and their respective controls. For each region, we tested whether individuals with a deletion had significantly higher (or lower) similarity to the other deletion FC-signatures than their respective controls using a Mann–Whitney $U$ test. We reported significant group differences after FDR correction ($q < 0.05$) for the number of regions (64).

**Aligning gene expression maps and functional parcellation.** To investigate the transcriptomic relationship of altered FC in each deletion, we aligned gene expression values in the adult human brain from the Allen Human Brain Atlas (AHBA) dataset[16] to the MIST64 brain parcellation following previously published guidelines for probe-to-gene mappings and intensity-based filtering[71] and adapting the abagen toolbox[72]. We normalized expression values within each brain sample across genes for each of the 6 donors and then for each gene across samples for each donor using a scaled robust sigmoid normalization[71]. We computed the mean of the normalized values of all samples encompassed within each functional region of the MIST64. This was performed for each donor and then averaged across donors. A leave-one-donor out sensitivity analysis generated six expression maps. The principal components of these six expression maps were highly correlated (average Pearson correlation of 0.993). The same high correlation was observed for the differential Stability score (average Pearson correlation of 0.987).

Normalized gene expression value was available for each of the 15663 genes and for each of the 64 functional brain regions.

**Gene expression analyses.** For all analyses, we used a dataset including one expression value per gene and per functional region. Expression values were associated with the average connectivity alteration of the corresponding regions (specifically, we computed the mean of all 64 beta values of each region, a measure of global connectivity).

PLSR method was used to investigate the association between spatial patterns of gene expression (of the 37 and 24 genes encompassed in the 22q11.2 and 16p11.2 genomic loci) and the 16p11.2 and 22q11.2 FC signatures. PLSR is a multivariate approach, which has previously been applied to investigate the relationship between neuroimaging phenotypes and spatial patterns of gene expression[73–76]. PLSR was performed separately for 16p11.2 and 22q11.2 genes. Components defined by PLSR were the linear combinations of the weighted gene expression scores (predictor variables) that most strongly correlated with FC-signatures of deletions (response variables). To assess significance, we recomputed PLSR using 5000 null FC-signature maps and counted the number of times the explained variance was higher than the original observation. Null FC-signatures were obtained by computing 5000 times the contrast between CNVs and controls after label shuffling for 16p11.2 and 22q11.2 separately. To investigate the association between FC alterations and expression patterns of individual genes, we computed Pearson correlations. The null distribution was defined by the same 5000 random FC-signatures described above.

To test the specificity of the relationship between gene expression and FC, we randomly sampled 10,000 gene sets ($n = 24$ for 16p11.2 genes and $n = 37$ for 22q11.2 genes) from 15,633 genes and re-computed the PLSR 10,000 times. The explained variance ($R$-squared) was used as test-statistics for the null distribution, and the p-value was calculated as the number of times the explained variance of the random gene-set exceeded the variance explained by 16p11.2 or 22q11.2 genes. A similar approach was performed for the individual gene correlations using median correlation as test-statistics.

**Statistics and reproducibility.** Statistical analyses of fMRI connectivity have been technically replicated by three authors, in Python (C.A.M. and S.G.W.U.) and R (C.S.).

**Reporting summary.** Further information on research design is available in the Nature Research Reporting Summary linked to this article.

## Data availability

ABIDE1, COBRE, ADHD200, 16p11.2 SVIP data are publicly available: http://fcon_1000. projects.nitrc.org/indi/abide/abide_I.html, http://schizconnect.org/queries/new, http://fcon_1000.projects.nitrc.org/indi/adhd200/, https://www.sfari.org/funded-project/simons-variation-in-individuals-project-simons-vip/. Around 1/3 of the SZ data (172 out of 484 subjects) can't be shared as participants did not consent to that at the time. For the 22q11.2 sample, raw data are currently available by request from the PI. Beta maps from all Connectome wide association studies (16p11.2 deletion and duplication, 22q11.2 deletion and duplication, ASD, SZ, and ADHD) performed in this study are available in the Supplementary Data 1 and on GitHub. Expression data used in this study are also available in Supplementary Data 4.

## Code availability

The processing scripts and custom analysis software used in this work are available in a publicly accessible GitHub repository with instructions on how to set up a similar computation environment and with examples of key visualizations in the paper: https://github.com/surchs/Neuropsychiatric_CNV_code_supplement.

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

## Acknowledgements

This research was supported by Calcul Quebec (http://www.calculquebec.ca) and Compute Canada (http://www.computecanada.ca), the Brain Canada Multi investigator research initiative (MIRI), funds from the Institute of Data Valorization (IVADO). S.J. is a recipient of a Canada Research Chair in neurodevelopmental disorders, and a chair from the Jeanne et Jean Louis Levesque Foundation. C.S. is supported by a fellowship from the Institute for Data Valorization. Kuldeep Kumar is supported by The Institute of Data Valorization (IVADO) Postdoctoral Fellowship program, through the Canada First Research Excellence Fund. This work was supported by a grant from the Brain Canada Multi-Investigator initiative (S.J.) and a grant from The Canadian Institutes of Health Research (S.J.). Dr P. Bellec is a fellow ("Chercheur boursier Junior 2") of the "Fonds de recherche du Québec—Santé", Data preprocessing and analyses were supported in part by the Courtois foundation (P.B.). We are grateful to all of the families at the participating Simons Simplex Collection (SSC) sites, as well as the principal investigators (A. Beaudet, R. Bernier, J. Constantino, E. Cook, E. Fombonne, D. Geschwind, R. Goin-Kochel, E. Hanson, D. Grice, A. Klin, D. Ledbetter, C. Lord, C. Martin, D. Martin, R. Maxim, J. Miles, O. Ousley, K. Pelphrey, B. Peterson, J. Piggot, C. Saulnier, M. State, W. Stone, J. Sutcliffe, C. Walsh, Z. Warren, E. Wijsman). We appreciate obtaining access to imaging and phenotypic data on SFARI Base. Approved researchers can obtain the Simons Variation in Individuals Project population dataset described in this study by applying at https://base.sfari.org. ABIDE I is supported by NIMH (K23MH087770), NIMH (R03MH096321), the Leon Levy Foundation, Joseph P. Healy, and the Stavros Niarchos Foundation. Data in the schizophrenia dataset were accessed through the SchizConnect platform (http://schizconnect.org). As such, the investigators within SchizConnect contributed to the design and implementation of SchizConnect and/or provided data but did not participate in the data analysis or writing of this report. Funding of the SchizConnect project was provided by NIMH cooperative agreement 1U01 MH097435. SchizConnect enabled access to the following data repository: The Collaborative Informatics and Neuroimaging Suite Data Exchange tool (COINS; http://coins.mrn.org/dx). Data from one study was collected at the Mind Research Network and funded by a Center of Biomedical Research Excellence (COBRE) grant, (5P20RR021938/P20GM103472) from the NIH to Dr. Vince Calhoun. Data from two other studies were obtained from the Mind Clinical Imaging Consortium Database. The MCIC project was supported by the Department of Energy under award number DE-FG02-08ER6458. MCIC is the result of the efforts of co-investigators from the University of Iowa, University of Minnesota, University of New Mexico, and Massachusetts General Hospital. Data from another study were obtained from the Neuromorphometry by Computer Algorithm Chicago (NMorphCH) dataset (http://nunda.northwestern.edu/nunda/data/projects/NMorphCH). As such, the investigators within NMorphCH contributed to the design and implementation of NMorphCH and/or provided data but did not participate in the data analysis or writing of this report. The NMorphCH project was funded by NIMH grant RO1 MH056584. Data from the UCLA cohort provided by C.E.B. (participants with 22q11.2 deletions or duplications and controls) was supported through grants from the NIH (U54EB020403), NIMH (R01MH085953, R01MH100900, R03MH105808), and the Simons Foundation (SFARI Explorer Award). Finally, data from another study were obtained through the OpenFMRI project (http://openfmri.org) from the Consortium for Neuropsychiatric Phenomics (CNP), which was supported by NIH Roadmap for Medical Research grants UL1-DE019580, RL1MH083268, RL1MH083269, RL1DA024853, RL1MH083270, RL1LM009833, PL1MH083271, and PL1NS062410.

## Author contributions

C.A.M., S.G.W.U., S.J., and P.B., designed the overall study and drafted the manuscript. C.A.M. and S.G.W.U. processed the data and performed all imaging analyses. P.O. preprocessed the SZ data and reviewed the manuscript. C.S. performed the statistical analyses and drafted the manuscript. K.K. and G.D. performed the gene expression analyses. A.L., E.D., P.O.Q., and G.H. contributed to the interpretation of the data and reviewed the manuscript. A.L., L.K., and C.E.B. provided the UCLA fMRI data. S.G., D.L., A.M., S.P., and E.S. provided the SZ data. C.E.B., A.C.E., and T.B. contributed to the interpretation of the data and drafted the manuscript. All authors provided feedback on the manuscript.

## Competing interests

The authors declare no competing interests.
