## [Peer Review File · Nature Communications]

Reviewers' comments:

Reviewer #1 (Remarks to the Author):

The paper by Moreau and colleagues examines the functional connectivity (FC) correlates of four CNVs that confer high risk for psychiatric disorders (16p11.2 and 22q11.2 deletion/duplication), particularly ASD and schizophrenia (Sz). Using an exploratory "connectome-wide association studies" (CWAS) approach, they find that gene dosage is associated with mirrored effects in the functional connectome, with 16p11.2 deletions and 22q11.2 duplications being associated with hyperconnectivity while 16p11.2 duplications and 22q11.2 deletions are associated with hypoconnectivity with larger effect sizes than seen for idiopathic ASD, Sz, and ADHD. The authors go on to show that the signatures of idiopathic Schizophrenia and ASD at the individual level are correlated with the group-level "signatures" of the four CNVs (primarily, 16p11.2 deletions/duplications and 22q11.2 deletions) and that there are regional relationships between symptom severity and the degree of similarity between individuals with idiopathic diagnoses and the CNV FC signatures. Finally, the authors show robust similarity between the gene co-expression patterns for the 16p11.2 and 22q11.2 deletions.

The paper presents an innovative set of analyses and addresses a novel and important research question about the similarity between the brain signatures of neuropsychiatric CNVs and those of idiopathic diagnoses. The proof-of-concept findings will be of broad interest to researchers working in the area of neuropsychiatric genetics and neuropsychiatry more generally. The paper is well written, the data and analyses are sound, the results are compelling, and the figures are excellent.

I have reviewed this paper once already for a different journal. My original review was very positive, though I highlighted some weaknesses, omissions, and issues for clarification. I am pleased to see that these concerns have been addressed by the authors in this version. In particular, the authors have improved their description of the participant samples and have clarified the analysis of similarities between CNV signatures and the idiopathic diagnoses at the individual-level, adding an explanatory schematic (Fig. 2). In addition, they have made changes to address concerns about sample size and gender distribution as well as the presence of psychiatric diagnoses amongst CNV carriers. There are several welcome additions to the paper, such as the gene co-expression analysis. Below, I have outlined some minor remaining issues and clarifications that could be addressed to further improve the paper.

There are several welcome additions to the discussion. However, I still find the discussion somewhat lacking in terms of consideration of the results in the light of existing literature and in terms of outlining the implications of the findings. For example, the pattern of underconnectivity obtained for ASD is different from the more mixed pattern reported in the literature and referred to in the introduction (e.g., Holiga et al.). Given that the data analyzed here (from the ABIDE sample) have already been reported on (by e.g., Di Martino et al., 2014), can the authors explain the discrepancy? Another point that merits elaboration in the discussion is the contrast between the overall categorical diagnostic effects and the individual-level correlations with CNV patterns – for example, does this suggest the potential presence of genetic subgroups within the idiopathic cohorts with more diverse connectopathies? On the other hand, given the direct comparison of CNV carriers and idiopathic groups, the prevalence of the same psychiatric diagnoses amongst CNV carriers (e.g., half of the 22q11.2 deletion carriers have a diagnosis of ASD, per Table 1) seems an important issue that should be considered in the discussion.

Supplementary Figure 8 shows that there is considerable variation in motion between cohorts, and that a very substantial proportions of frames have been removed ("scrubbed") from the analysis for many participants. Further, the minimum number of frames (40) required for inclusion in the study is very low. Importantly, there appear to be very significant case-control cohort differences in the extent of scrubbing. Are the authors confident that the results are robust to the choice of motion mitigation strategy – that is, that the same results would be obtained if a different

approach than “scrubbing” had been used such as global signal regression or participant elimination? In the context of group comparisons, what is the impact of one group having significantly fewer frames than the other from which to compute FC?

Page 5 – the expression “mirror effects” appears here for the first time (after the abstract). The meaning of this phrase may not be clear to all readers and should be explained upon its first appearance.

Table 1 – the legend doesn’t explain the meaning of the right-most columns indicating diagnoses.

Clare Kelly

Reviewer #2 (Remarks to the Author):

The study describes a connectome-wide association study based on functional connectivity. The novel part of the study is comparing carriers of pathogenic CNVs (16p11.2 and 22q11.2) and idiopathic psychiatric conditions (autism, schizophrenia and ADHD). But there are previous studies that did analyzed functional connectivity in 22q11.2 and 16p11.2 deletion carriers – although not by CWAS. Deletions and duplications of the above CNVs showed opposite effects, which strengthen the reliability of the results, with effect size larger compared to the psychiatric conditions. They also analyses gene co-expression as a way to understand the similarities between the CNVs.

General comment:

Overall I found the study interesting, but because of its descriptive nature, the paper does not provides new understanding and mechanisms for psychiatric disorders or for the effect of the CNVs. Giving the large number of tests and potential confoundings it is not clear that the findings are directly related to the conditions.

Specific comments:

1. In this study the CWAS was conducted using linear regression. I am not sure that this is the best way to analyses the data. Most common methods are based on general linear model (GLM), but there are more advanced and robust methods to analyses the data:

Gong W, Cheng F, Rolls ET, Lo CY, Huang CC, Tsai SJ, Yang AC, Lin CP, Feng J. A powerful and efficient multivariate approach for voxel-level connectome-wide association studies. *NeuroImage*. 2019 Mar 1;188:628-41.

2. In this study they found overall under-connectivity for ASD, but I am not sure what the explanation is for the inconsistencies with previous studies.

Reviewer #3 (Remarks to the Author):

This premise of this study by Moreau is an important one: how do common CNVs associated with neuropsychiatric disorders correlate with functional connectivity measures as assessed by fMRI? To address this question, the authors assembled data from a large cohort of individuals with either deletion or duplication of 16p11.2 or 22q11.2 together with a cohort of individuals with either idiopathic autism or schizophrenia (and matched controls for both).

The authors find that at a network level, 16p11.2 has a mirror effect but 22q11.2 does not. One of the most interesting findings is the integration of symptom data with the FC signatures. In

this case, similarity between idiopathic ASD and CNV deletions correspond to particular connections and ADOS/FSIQ scores, whereas no such similarities were found for schizophrenia data.

While this manuscript has some results that might be informative for the field, there are some issues that should be addressed to improve the presentation of the results.

Major issues:

1) In general, each of the main results appears to be of general interest for understanding the biology of these disorders, however, the authors do not link the finding together in a cohesive manner. It is still not clear to me how specific CNV deletions or duplications are related to brain properties. I am only left with the knowledge that these are still somehow related to "abnormal" brain function, that there are similarities and differences, but a clearly defined mechanism or hypothesis is not generated.

2) To understand which regions are responsible for dysconnectivity, the authors find the regions with preserved connectivity in CNV and idiopathic conditions but not in control subjects. Since delineating the regions important for this phenomenon is a crucial part of any in-depth future study, it would be much more comprehensive if the authors grouped the regions based on altered connectivity w

Reviewer 1

Preamble

The paper presents an innovative set of analyses and addresses a novel and important research question about the similarity between the brain signatures of neuropsychiatric CNVs and those of idiopathic diagnoses. The proof-of-concept findings will be of broad interest to researchers working in the area of neuropsychiatric genetics and neuropsychiatry more generally. The paper is well written, the data and analyses are sound, the results are compelling, and the figures are excellent.

I have reviewed this paper once already for a different journal. My original review was very positive, though I highlighted some weaknesses, omissions, and issues for clarification. I am pleased to see that these concerns have been addressed by the authors in this version. In particular, the authors have improved their description of the participant samples and have clarified the analysis of similarities between CNV signatures and the idiopathic diagnoses at the individual-level, adding an explanatory schematic (Fig. 2). In addition, they have made changes to address concerns about sample size and gender distribution as well as the presence of psychiatric diagnoses amongst CNV carriers. There are several welcome additions to the paper, such as the gene co-expression analysis. Below, I have outlined some minor remaining issues and clarifications that could be addressed to further improve the paper.

There are several welcome additions to the discussion.

Question 1

However, I still find the discussion somewhat lacking in terms of consideration of the results in the light of existing literature and in terms of outlining the implications of the findings. For example, the pattern of underconnectivity obtained for ASD is different from the more mixed pattern reported in the literature and referred to in the introduction (e.g., Holiga et al.). Given that the data analyzed here (from the ABIDE sample) have already been reported on (e.g., Di Martino et al., 2014), can the authors explain the discrepancy?

Response: We fully agree that it is important to discuss the consistency of our findings with previous studies on ASD and schizophrenia. As suggested by the reviewer, we compared our findings on ASD with the paper by Di Martino¹ that also used data from the ABIDE 1 sample. Despite preprocessing and analytical differences in the work of Di Martino, the pattern of ASD associated FC-alterations is highly consistent. Our findings of general underconnectivity with overconnectivity of cortico-subcortical connections between the thalamus and

sensorimotor areas (Supplementary Figure 4.a-d) are in line with those reported by Di Martino, and other studies²⁻⁴ (eg. the “Generalizability and reproducibility of functional connectivity in autism” by King et al).

Work by Holiga and colleagues⁵ reports findings of prefrontal and parietal overconnectivity, and sensory-motor underconnectivity in autism. A direct comparison with these findings is more challenging, both because of differences of methodology (data in Holiga’s work are corrected for global signal during preprocessing and analyzed with respect to degree centrality) and analytic focus. Holiga’s work targets reproducibility of degree centrality findings, and FC alterations are reported for areas within a mask of altered degree centrality. Authors report reproducible alterations of degree centrality across datasets that include ABIDE using a mask based on FC alterations observed in the EU-AIMS data that is not currently available to the research community.

Based on our new review of the literature, we modified the introduction:

“... Studies applying this analytical strategy in ASD have repeatedly shown patterns of widespread under-connectivity with the exception of overconnectivity in cortico-subcortical connections, particularly involving the thalamus^{1,2}...”

We also added the following section in the discussion:

“...Reproducibility of rs-fMRI in psychiatry has been challenging. However, when studies using similar analytical strategies are compared, there are consistent results. In SZ and ASD, a global decrease in FC has been reported by most studies except for those adjusting for global signal^{2,6}. Increased thalamocortical connectivity is also repeatedly reported in both conditions^{2,7,8}. These previous findings are consistent with our results (see Supplementary Results). The 22q11.2 deletion FC signature is consistent with previous works on 22q11.2 FC alterations that showed 1) underconnectivity of the DMN^{9,10}, 2) thalamocortical overconnectivity and underconnectivity involving the hippocampus¹¹. The only rs-fMRI study previously published for the 16p11.2 deletion focused on the dmPFC¹². Using the same approach and regressing global signal, we also found underconnectivity of the dmPFC with the same set of regions. This highlights the fact that many seemingly discrepant results can be reconciled once methodologies are aligned...”

Finally, we added this Supplementary figure 4.

Supplementary Figure 4: Effects of ASD and SZ on FC before and after adjustment for global signal. The adjustment removes the mean shift (global underconnectivity observed in ASD and SZ). The alterations of each region and network relative to one another remain similar before and after adjustment and both FC-signatures (beta maps) are highly correlated ($r=0.98$). This also demonstrates that ASD and SZ are associated with a mean shift in connectivity as well as a reorganisation of networks relative to one another.

Question 2

Another point that merits elaboration in the discussion is the contrast between the overall categorical diagnostic effects and the individual-level correlations with CNV patterns – for example, does this suggest the potential presence of genetic subgroups within the idiopathic cohorts with more diverse connectopathies?

Response: It is highly unlikely that subgroups of CNV carriers within the idiopathic cohorts are driving the CNV-idiopathic similarities observed in our study. Indeed, 22q11.2 and 16p11.2 CNVs are observed in less than 1% of autism or schizophrenia cohorts ¹³.

We added the following sentences in the limitations :

“There is no available genetic data for any of the three idiopathic cohorts. However, the frequency of 16p11.2 and 22q11.2 CNVs in ASD or SZ is < 1% ^{13,14}. This suggests that the observed FC similarities between CNVs and ASD or SZ, are driven by other factors.”

Question 3

On the other hand, given the direct comparison of CNV carriers and idiopathic groups, the prevalence of the same psychiatric diagnoses amongst CNV carriers (e.g., half of the 22q11.2 deletion carriers have a diagnosis of ASD, per Table 1) seems an important issue that should be considered in the discussion.

Response: This is indeed an important issue to consider. Previous studies including some from our lab ¹⁵ have demonstrated that the main structural neuroimaging alterations associated with CNVs were not influenced by the different psychiatric diagnoses present in CNV carriers ^{15,16}. This suggests that the neuroimaging alterations reflect the biological risk associated with the CNV and not the final diagnosis which results from the sum of CNV effects + additional factors.

To address this comment, we performed a sensitivity analysis to understand if the presence of a psychiatric diagnosis impacts the CNV FC signature. We compared the FC signatures of 22q11.2 deletion carriers with and without ASD. Both FC signatures showed the same global underconnectivity (Supplemental Figure 1) and were strongly correlated with the one presented in the manuscript despite the smaller sample size ($r=0.83$ to 0.90). The same sensitivity analysis also showed that ADHD diagnosis had no detectable influence on the 22q11.2 deletion FC signature.

We performed the similarity analyses between 22q11.2 deletion profiles and idiopathic conditions after exclusion of 22q11.2 subjects with either ASD or ADHD. Results in the new supplemental Figure 1 (below) show that the same regions are driving similarities with 22q11.2 FC profiles irrespective of diagnoses.

To conclude, although there is likely an effect of psychiatric diagnosis on CNV connectivity profiles, we expect those effect sizes to be small (comparable to those observed in cases control studies of ASD and SZ). Therefore, it would require CNVs sample sizes in the hundreds to observe the effects of ascertainment and psychiatric diagnoses in CNV carriers.

We added a section on this point in the legend of table 1:

“...Sensitivity analysis also showed that the FC-signature of 22q11.2 deletions is not influenced by a diagnosis of ASD or ADHD...”

In the results :

“... Sensitivity analysis showed that the same regions are driving similarities with SZ and ASD irrespective of psychiatric diagnoses in 22q11.2 deletion carriers (see supplemental Figure 1)....”

And in the discussion :

“...Those include sex bias, which is present across all 3 psychiatric cohorts, age differences in the 16p11.2 deletion group, diagnosis of ASD and ADHD in 22q11.2 deletion carriers, and...”

We added the following figure in Supplementary results:

Supplemental Figure 1: Sensitivity analyses psychiatric diagnoses in 22q11.2 deletion

Legend: a) Density plots show the distribution of beta coefficients of CWAS contrasting controls and subgroups of 22q11.2 deletion (with and without diagnoses, in orange and green respectively). The full 22q11.2 sample is in blue. Under-connectivity is present in all 22q11.2 subgroups irrespective of diagnosis. b) Similarities between 22q11.2 regional FC-signature and individuals with ASD and SZ. The same regions showed similarities in all 22q11.2 subgroups irrespective of psychiatric diagnosis, despite the 2-fold decrease in sample size. Maps are thresholded maps (FDR). Color scale represents the similarity effect size (Mann Whitney, rank biserial correlation).

Question 4

Supplementary Figure 8 shows that there is considerable variation in motion between cohorts and that a very substantial proportions of frames have been removed (“scrubbed”) from the analysis for many participants. Further, the minimum number of frames (40) required for inclusion in the study is very low. Importantly, there appear to be very significant case-control cohort differences in the extent of scrubbing.

Response:

Motion is an essential confounding factor in populations with neurodevelopmental disorders. We have carefully investigated the potential effects of motion and scrubbing throughout the analyses. As requested by the reviewer, we performed additional sensitivity analyses by excluding all individuals with less than 60 frames (n=4 deletion carriers, 4 duplication carriers, 10 controls). The CWAS results are very close to those obtained with all participants. FC-signatures before and after exclusion showed an $r=0.94$ and $r=0.89$ for 16p11.2 deletions and duplications respectively.

We also analyzed additional deletions (n=16), and duplications (n=5) and controls (n=1300) from 4 additional sites including the UK Biobank. All individuals had >60 frames. The new FC-signatures profiles were correlated with the initial ones ($r=0.79$ et $r=0.87$ for deletions and duplications) and demonstrated similar overconnectivity and thalamo-sensory disturbance despite different ascertainment, and noise introduced by additional sites. The 16p11.2 FC-signature obtained with this larger sample shows the same similarities with ASD and SZ and the thalamus remains a central region driving similarities with both conditions. We did not include these new data in the manuscript, because they do not change the results and conclusions of the study and would require updating every single downstream analysis throughout the main manuscript and supplemental file.

Of note, we did not perform this sensitivity analysis on the 22q11.2 data since all individuals had >60 frames.

We added the following information in the results:

“...A sensitivity analysis showed that results are unaffected by differences in age distribution between deletions and control groups as well as the number of remaining frames available after scrubbing (see Supplementary Results)...”

As well as in the Supplementary Results:

“Sensitivity analysis on the number of remaining frames in 16p11.2 deletion carriers

We performed a sensitivity analysis by excluding all individuals with less than 60 frames (n=4 deletion carriers, 4 duplication carriers, 10 controls). The CWAS results were very close to those obtained with all participants. FC-signatures before and after exclusion showed an $r=0.94$ and $r=0.89$ for 16p11.2 deletions and duplications respectively. “

Question 5

Are the authors confident that the results are robust to the choice of motion mitigation strategy – that is, that the same results would be obtained if a different approach than “scrubbing” had been used such as global signal regression or participant elimination? In the context of group comparisons, what is the impact of one group having significantly fewer frames than the other from which to compute FC?:

Response :

We also performed additional analyses by regressing for the global signal. FC-signatures of deletions and duplication before and after this adjustment were highly correlated: $r=0.90$ and $r=0.85$ respectively. Of note, the GSR centred the beta maps (eg. 16p11.2 deletions showed a similar proportion of under and overconnectivity alterations after global signal regression compared to general overconnectivity before adjustment but the ranking of alterations remains identical). We also performed additional analyses by regressing for the global signal in the idiopathic psychiatric conditions. FC-signatures before and after this adjustment were also highly correlated ($r>0.97$).

Overall, these sensitivity procedures together with the analysis of novel, independent data show that the connectome profiles of CNV carriers are robust and not secondary to motion-related artefacts.

Question 6

Page 5 – the expression “mirror effects” appears here for the first time (after the abstract). The meaning of this phrase may not be clear to all readers and should be explained upon its first appearance.

Response: We gave a definition of mirror effect in the introduction :

“Gene dosage (deletions and duplications) affect the same neuroimaging measures in opposite directions (mirror effect).”

Question 7

Table 1 – the legend doesn’t explain the meaning of the right-most columns indicating diagnoses.

Response: The right-most column represents the number of each diagnosis in each group. Subjects can have multiple psychiatric diagnoses.

We added the following legend :

“...Columns ASD, SZ, and ADHD represent the number of subjects with those diagnoses. One subject may have several diagnoses. For example, 9 subjects with ASD also have an ADHD diagnosis. ...”

Reviewer 2

Preamble

The study describes a connectome-wide association study based on functional connectivity. The novel part of the study is comparing carriers of pathogenic CNVs (16p11.2 and 22q11.2) and idiopathic psychiatric conditions (autism, schizophrenia and ADHD). But there are previous studies that did analyzed functional connectivity in 22q11.2 and 16p11.2 deletion carriers – although not by CWAS. Deletions and duplications of the above CNVs showed opposite effects, which strengthen the reliability of the results, with effect size larger compared to the psychiatric conditions. They also analyses gene co-expression as a way to understand the similarities between the CNVs.

Discussion

Question 8

Overall I found the study interesting, but because of its descriptive nature, the paper does not provides new understanding and mechanisms for psychiatric disorders or for the effect of the CNVs.

Response: This is a fundamental question. All human genetic studies are based on statistical associations and mechanisms can only be inferred but not directly tested. This paper provides a full characterization of the connectivity alterations in CNVs carriers. Subsets of these alterations are similar to those observed in idiopathic autism or schizophrenia who do not carry CNVs. In particular, the overconnectivity between the thalamus and sensory-motor, auditory and visual networks are altered across CNVs and psychiatric conditions.

Sensory processing and perceptual dysfunction is a core feature in SZ and ASD ¹⁷. Auditory and visual hallucinations are experienced by more than 69% and 27% of patients with SZ ^{18,19}. Individuals with autism present several visual task impairment such as gestalt perception and discrimination of visual motion ²⁰. They also present disturbances in auditory and tactile discrimination tasks. Hippolyte and colleagues showed that deletion carriers present severe impairments of phonology ²¹. Deletion at the 22q11.2 locus is also strongly associated with visual and auditory deficits ²²⁻²⁴. We, therefore, infer that a general thalamo-sensory disturbance may be central across psychiatric diagnoses and genomic mutations. This in itself is novel and may explain why so many genomic variants lead to the same psychiatric conditions. Animal studies may help understand the convergence of genomic mutations on large scale brain alterations.

We modified the discussion to highlight this mechanistic hypothesis :

“Among the regions, results highlighted overconnectivity between the thalamus and sensory-motor, auditory and visual networks as a common alteration across CNVs and individuals with idiopathic autism or schizophrenia who do not carry CNVs. This is in line with recent rs-fMRI studies performed across psychiatric illnesses ²⁵. Perceptual dysfunction are core features of SZ and ASD ¹⁷. Those include auditory and visual hallucinations in SZ ^{18,19}, impairments in gestalt visual perception and discrimination of visual motion ²⁰, disturbances in auditory and tactile discrimination in Autism. Impairments in phonology ²¹, as well as visual and auditory deficits, have also been demonstrated in 16p11.2 and 22q11.2 deletion carriers ²²⁻²⁴. A general thalamo-sensory disturbance may, therefore, be central across genomic mutations and psychiatric diagnoses. Further studies are required to investigate genome-wide, the genetic determinants of thalamo-sensory disturbance. Because it appears ubiquitous across conditions, the genetic basis is likely to be very broad. ”

Question 9

Giving the large number of tests and potential confoundings it is not clear that the findings are directly related to the conditions.

Response: We performed multiple sensitivity analyses and showed that the effects of the CNVs are robust and not secondary to confounding factors. This includes investigating the effect of sex (cf supplementary methods), the effect of psychiatric diagnosis in CNV carriers (cf. response to comment 3 of reviewer# 1), the effect of motion (cf. response to comment 4 of reviewer# 1), the effect of the number of remaining frames (cf. response to comment 5 of reviewer# 1). We also analyzed new data on CNV carriers that confirm the robustness of the FC-signatures (cf. response to comment 4 of reviewer# 1).

Question 10

In this study they found overall under-connectivity for ASD, but I am not sure what the explanation is for the inconsistencies with previous studies.

Response: The reviewer 1 also asked to compare findings with previous reports. We provided a detailed comparison of our findings on ASD with previous reports and show highly correlated results. We also provided a comparison with SZ, which also shows highly consistent results. (cf. reviewer 1, comment#1).

We added the following section in the discussion:

“...Reproducibility of rs-fMRI in psychiatry has been challenging. However, when studies using similar analytical strategies are compared, there are consistent results. In SZ and ASD, global decrease in FC has been reported by most studies except for those adjusting for global signal^{2,6}. Increased thalamocortical connectivity is also repeatedly reported in both conditions^{2,7,8}. These previous findings are consistent with our results (see Supplementary Results). The 22q11.2 deletion FC signature is consistent with previous works on 22q11.2 FC alterations that showed 1) underconnectivity of the DMN^{9,10}, 2) thalamocortical overconnectivity and underconnectivity involving the hippocampus¹¹. The only rs-fMRI study previously published for the 16p11.2 deletion focused on the dmPFC¹². Using the same approach and regressing global signal, we also found underconnectivity of the dmPFC with the same set of regions. This highlights the fact that many seemingly discrepant results can be reconciled once methodologies are aligned...”

Question 11

In this study the CWAS was conducted using linear regression. I am not sure that this is the best way to analyse the data. Most common methods are based on general linear model (GLM), but there are more advanced and robust methods to analyses the data : Gong W, Cheng F, Rolls ET, Lo CY, Huang CC, Tsai SJ, Yang AC, Lin CP, Feng J. A powerful and efficient multivariate approach for voxel-level connectome-wide association studies. NeuroImage. 2019 Mar 1;188:628-41.

Response:

The reviewer suggested that a voxel-wise multivariate approach may represent a more powerful method to analyze the resting-state data, specifically citing the work²⁶. The Gong et al. (2019) technique entails two ideas: reducing the dimensionality of connectome data through a kernel principal component analysis, and using an adaptive regression technique instead of the general linear model (GLM) to select the optimal resolution. Our technique actually follows a similar philosophy: Bellec et al.²⁷ proposed a multi-resolution dimensionality reduction technique based on ensemble clustering²⁸ applied on an independent dataset to create a reference atlas²⁹. Multiple comparisons are corrected using false-discovery rate (FDR³⁰). Bellec et al.²⁷ demonstrated using extensive simulation and empirical data that (1) the FDR algorithm precisely controls the effective FDR at nominal value, i.e. the statistical technique works as expected; (2) the resolution of the atlas achieves a critical trade-off between statistical power (sensitivity, lower resolution is better) and the match between parcel-based and voxel-based results (compression quality, higher resolution is better). We selected the scale for the present analysis based on this prior work. The reviewer claims that Gong et al.²⁶ achieves more statistical power, but no comparison was made with our algorithm in this reference. Because the philosophy of both techniques is

somewhat similar (data-driven dimension reduction, dimensionality optimization), it is not clear to the authors that substantial gains could actually be achieved.

We attempted to evaluate the procedure proposed by Gong et al., 2019 which is available through a public code (<https://github.com/weikanggong/sKPCR>), at a beta stage of development. A listed requirement for the sKPCR library is “Your CPU memory should be enough to put all your rfMRI data into it.” which would only be possible on select servers from the high-performance computing facility our lab has access to. In the documentation, the authors suggested using a separate library (<https://github.com/weikanggong/BWAS>) which does not have the same memory requirement, so we started benchmarking that other algorithm. However, both applications (sKPCR and BWAS) do not let users flexibly select a subset of subjects in their analysis. We thus had to implement a wrapper in order to try and replicate our analysis (https://github.com/simexp/bwas_wrapper). We ran a pilot test on 30 subjects (https://github.com/SIMEXP/bwas_wrapper/blob/master/bwas_demo.py) which took several hours to run on a 64 cores server, and eventually crashed because of a missing dependency on Matplotlib’s jpg plugin. This dependency was not listed in the requirements of the library. At this stage we abandoned our experiment, as computation time with hundreds of subjects would be absolutely prohibitive.

In summary, our approach has several key advantages to the approach suggested by the reviewer: (1) our brain parcels are carefully labeled for interpretability ²⁹ and publicly available in the popular Nilearn library (<https://nilearn.github.io/>), which is very well documented and tested, and straightforward to install; (2) the FDR algorithm is widely known and used, has been carefully validated for our application, and high-quality implementations are available in all major languages; (3) because of the selected low resolution, the analysis runs in a matter of seconds on a low-end machine, which greatly facilitates reproducibility at a practical level; (4) we were able to implement a similarity analysis between CNV signatures and individual connectivity profiles in idiopathic populations because of the low number of brain parcels. Increasing the number of parcels would not only increase the computational load, but also dramatically inflate the apparent effect size, an issue well documented in voxel-based analyses ³¹. As we stated above, the only advantage mentioned by the reviewer (increased statistical power) is actually not established and, if true, would not outweigh the advantages listed above.

Reviewer 3

This premise of this study by Moreau is an important one: how do common CNVs associated with neuropsychiatric disorders correlate with functional connectivity measures as assessed by fMRI? To address this question, the authors assembled data from a large cohort of individuals with either deletion or duplication of 16p11.2 or 22q11.2 together with a cohort of individuals with either idiopathic autism or schizophrenia (and matched controls for both).

The authors find that at a network level, 16p11.2 has a mirror effect but 22q11.2 does not.

One of the most interesting findings is the integration of symptom data with the FC signatures. In this case, similarity between idiopathic ASD and CNV deletions correspond to particular connections and ADOS/FSIQ scores, whereas no such similarities were found for schizophrenia data.

While this manuscript has some results that might be informative for the field, there are some issues that should be addressed to improve the presentation of the results.

Question 12

In general, each of the main results appears to be of general interest for understanding the biology of these disorders, however, the authors do not link the finding together in a cohesive manner. It is still not clear to me how specific CNV deletions or duplications are related to brain properties. I am only left with the knowledge that these are still somehow related to “abnormal” brain function, that there are similarities and differences, but a clearly defined mechanism or hypothesis is not generated.

Response: The reviewer is absolutely correct and so far, results do not identify relationships between specific CNVs and brain properties. This is consistent with the clinical studies showing that many different CNVs confer risk for a similar range of psychiatric phenotypes³². In fact, there is mounting evidence in the genetic literature that genes may not have specific effects on brain properties. As an example, studies show that common variation in every megabase in the genome has a 75% probability of increasing risk for SZ³³. Studies have also estimated that any 1MB deletion (containing coding genes) across the genome increases risk for ASD. Such results are puzzling and question the nature of the relationship between genomic variants and higher-order brain phenotypes such as large-scale connectivity and behaviour.

The main brain property affected across both CNVs and idiopathic conditions is the connectivity between thalamus and cortical sensory networks which is also correlated to ASD severity. We added the following section in the discussion:

“Among the regions, results highlighted overconnectivity between the thalamus and sensory-motor, auditory and visual networks as a common alteration across CNVs and individuals with idiopathic autism or schizophrenia who do not carry CNVs. This is in line with recent rs-fMRI studies performed across psychiatric illnesses²⁵. Perceptual dysfunction are core features of SZ and ASD¹⁷. Those include auditory and visual hallucinations in SZ^{18,19}, impairments in gestalt visual perception and discrimination of visual motion²⁰, disturbances in auditory and tactile discrimination in Autism. Impairments in phonology²¹, as well as visual and auditory deficits, have also been demonstrated in 16p11.2 and 22q11.2 deletion carriers²²⁻²⁴. A general thalamo-sensory disturbance may, therefore, be central across genomic mutations and psychiatric diagnoses. Further studies are required to investigate genome-wide, the genetic determinants of thalamo-sensory disturbance. Because it appears ubiquitous across conditions, the genetic basis is likely to be very broad.”

We added the following section in the conclusion

“Thalamo-sensory disturbance may represent one dimension central across genomic mutations and psychiatric diagnoses. The redundant associations observed, genome-wide, between gene expression and connectivity may explain similarities across genomic variants and idiopathic conditions and the fact that many CNVs affect a similar range of neuropsychiatric symptoms.”

Question 13

To understand which regions are responsible for dysconnectivity, the authors find the regions with preserved connectivity in CNV and idiopathic conditions but not in control subjects. Since delineating the regions important for this phenomenon is a crucial part of any in-depth future study, it would be much more comprehensive if the authors grouped the regions based on altered connectivity with respect to control subjects (i.e. separately calculate FC for SCZ, ASD, CNV, CTL per region and compare). Among many other things, this would tell us:

- are there any regions altered in SCZ or ASD but not in individuals with CNVs?
- are there regions altered in SCZ or ASD that are more severely affected with CNVs?
- which regions are the most affected overall?

Response: We assume that the reviewer meant “...the authors find the regions with altered (not preserved) connectivity in CNV and idiopathic conditions...”.

- Regions altered in SZ and ASD but not in individuals with CNVs are the cerebellum Crus-1, the Cerebellum-7b, the right inferior frontal sulcus, the left superior frontal gyrus anterior (Figure A.a).
- Regions altered in SZ and ASD that showed higher percentage discovery in CNVs (top 10, Figure A.b) include the perigenual anterior cingulate cortex, the lateral frontal pole, the caudate nucleus, the dorsal anterior cingulate cortex, the anterior medial frontal gyrus, the inferior temporal gyrus, the putamen, the right cerebellum crus 2, the lateral occipital gyrus, the anterior lateral visual network.
- Regions with the highest percentage discovery overall (Figure A.c) are the dorsal anterior cingulate cortex, the perigenual anterior cingulate cortex, the dorsal posterior insula, the somatomotor network. the ventral anterior insula and posterior insular sulcus, the temporal pole, the ventral posterior insula, the anterior medial temporal gyrus, the inferior marginal sulcus.

Figure A Legend: a) Regions altered in SCZ and ASD but not in individuals with CNVs, b) Regions altered in SZ and ASD that showed higher percentage discovery in CNVs, c) Regions with the highest percentage discovery overall.

We did not include these brain maps in the main manuscript because demonstrating an absence of association is highly sensitive to sample size. Indeed, a larger sample of CNV carriers will most likely identify additional connectivity alterations CNV carriers and idiopathic conditions.

Question 14

The authors attempt to integrate gene expression data from the Allen Human Brain Atlas for the genes that are found within the CNV blocks. However, these results are ultimately not integrated with the CNV-specific FC findings in a substantial way nor are they linked to the symptom findings. In fact, the gene co-expression analysis could be performed without the knowledge gained from this paper regarding the FC. For example, it is not clear how the 34 brain regions used to look at gene expression were even selected.

Response: As the reviewer accurately pointed out, we did not directly integrate the gene expression data with the CNV-specific FC findings. To address this important comment, we completely re-analyzed the data and rewrote this section:

- 1) We first remapped the gene expression data from Allen Human Brain Atlas onto the 64 MIST brain parcellation used throughout our study (the 34 regions of the Desikan atlas were initially selected because gene expression had previously been mapped onto this atlas by our collaborators).
- 2) We investigated the association between functional connectivity alteration profiles and gene expression.
- 3) We have included the code for reproducing the analysis reported in this section in the publicly shared Github repository (https://github.com/surchs/Neuropsychiatric_CNV_code_supplement.)

- The new result section reads as follow:

“Association between gene expression spatial patterns and deletion FC-signatures

We performed Partial Least Squares Regression (PLSR) to investigate the association between FC-signatures of each deletion and the expression patterns of 37 and 24 genes encompassed in the 22q11.2 and 16p11.2 genomic loci respectively. The 2 components required to reach a significant association explained 24.2% of the variance of the 16p11.2 deletion FC profile ($p=0.041$, 5000 random FC profiles). For the 22q11.2 deletion, either one or 2 components were significant ($p<0.0002$, 5000 random FC profiles). The 2 components explained 43.2% of the

variance of the 22q11.2 deletion FC signature. Similar PLSR analyses performed for each of the 64 regions showed that 18 and 32 regional FC-signatures were significantly associated (5000 random FC-signatures, FDR 64 regions) with spatial patterns of gene expression at the 16p11.2 and 22q11.2 loci respectively (Figure 5b-c). However, this relationship was not specific because 22q11.2 genes were also associated with n=20 regions of the 16p11.2 FC signature. Conversely, the 16p11.2 genes were associated with n=19 regions of the 22q11.2 FC signature (Figure 5.c, Supplementary Table S4.2).

To further investigate the low specificity of the connectivity/gene expression relationship, we tested the individual correlation (Pearson) of all 15663 genes with available expression data with deletion FC signatures (Figure 5d, e). Correlations with the 16p11.2 and 22q11.2 FC signatures were observed for 421 and 3883 genes respectively (5000 random FC-signatures). After genome-wide FDR correction, the expression of 1834 genes remained spatially correlated with 22q11.2 and none with the 16p11.2 deletion FC signature. The median correlation values for the n=24 16p11.2 genes and the 16p11.2 FC signature was not higher than the median correlation of 10000 randomly sampled gene sets of the same size (n=24, p=0.31). The same was true for 22q11.2 genes (n=37, p=0.36). However, both deletions were enriched in genes with correlations ranking higher than the genome-wide 98th percentile: *MVP* and *KIF22* showed correlations ($r_{MVP}=0.33$, $r_{KIF22}=0.26$) ranking at the 99.76th and 98.76th percentile genome-wide (p=0.03; null:10000 random gene sets). For 22q11.2, *AIFM3*, *TBX1* and *P2RX6* showed correlations in the 99.5, 98.84, 98.33th percentile (p=0.04).”

- We replaced the Figure 5 by the following figure:

“Figure 5. FC similarities between 16p11.2 and 22q11.2 and relationship with gene expression

a) 16p11.2 and 22q11.2 FC-signatures

b) 16p11.2 genes / 16p FC-signature

c) 22q11.2 genes / 22q FC-signature

d) CNVs genes / CNVs FC-signatures

e) 16p-genes 16p11.2 FC-signature

f) 22q-genes 22q11.2 FC-signature

Legend: (a) FC similarities between both deletions at the regional level. The values in the brain map represent the level of the FC similarity between deletions (rank biserial correlation, Mann Whitney test). The values are thresholded (FDR, 64 regions): 18 out of 64 regions are similar between deletions.

(b) Relationship between spatial patterns of gene expression within the 16p11.2 locus and regional FC signatures of the 16p11.2 deletion. A partial least square regression (PLSR) was conducted for each of the 64 regions. Maps are thresholded (FDR corrected for 64 regions) and color code represents the percentage of variance explained by gene expression using 2 components in the PLSR.

(c) The same analysis was conducted for 22q11.2 genes and the 22q11.2 deletion FC signature.

(b,c) Eleven regions overlapped across PLSR maps: thalamus, caudate, anterior insula and posterior insula sulcus, amygdala and hippocampus, cerebellum 9 and right crus-2, dorsal anterior cingulate, left inferior parietal lobule, medial posterior visual network, lateral posterior visual network, and dorsal visual network. Three (over the 18) regions identified in the between deletion similarity (a) analysis are also present in the gene

expression/FC-signature association maps (b,c): Thalamus, dorsal anterior cingulate and left inferior parietal lobule.

(d) Low specificity for the relationship between spatial patterns of gene expression and regional FC deletion signatures. In red: the 16p11.2 regional FC associated with the expression patterns of both the 16p11.2 and the 22q11.2 genes. In blue, the 22q11.2 regional FC is associated with the expression patterns of genes in both genomic loci. In purple, 7 regions were found with both deletion FC-signatures, and the expression patterns of genes encompassed in both genomic loci.

(e-f) Expression patterns of genes within and outside CNVs correlate with FC-signatures of 16p11.2 and 22q11.2 deletions. The light blue histogram represents the distribution of correlations for 15663 genes with available gene expression data from the AHBA. X-axis values : Pearson coefficients. Y-axis values: number of genes. Genes within the CNVs have font-size scaled based on p values. Dotted lines represent the 5th and 95th percentiles of the correlation distribution genome-wide.”

- We changed the method section in the manuscript:

“Gene expression analyses

We aligned the gene expression maps from AHBA to the MIST64 functional parcellation following previously published guidelines³⁴ and adapting the abagen toolbox³⁵ (supplementary methods). For all analyses, we used a dataset including 1 expression value per gene and per functional region. Expression values were associated with the average connectivity alteration of the corresponding regions (mean of all 64 beta values of each region).

PLSR method was used to investigate the association between spatial patterns of gene expression (of the 37 and 24 genes encompassed in the 22q11.2 and 16p11.2 genomic loci) and the 16p11.2 and 22q11.2 FC signatures. PLSR is a multivariate approach, which has previously been applied to investigate the relationship between neuroimaging phenotypes and spatial patterns of gene expression³⁶⁻³⁹. PLSR was performed separately for 16p11.2 and 22q11.2 genes. Components defined by PLSR were the linear combinations of the weighted gene expression scores (predictor variables) that most strongly correlated with FC-signatures of deletions (response variables). To assess significance, we recomputed PLSR using 5000 null FC-signature maps and counted the number of times the explained variance was higher than the original observation. Null FC-signatures were obtained by computing 5000 times the contrast between CNVs and controls after label shuffling for 16p11.2 and 22q11.2 separately.

To investigate the association between FC alterations and expression patterns of individual genes, we computed Pearson correlations. The null distribution was defined by the same 5000 random FC-signatures described above. To test the specificity of the relationship between gene expression and FC, we randomly sampled 10000 gene

sets (n=24 for 16p11.2 genes and n=37 for 22q11.2 genes) from 15633 genes and re-computed the PLSR 10000 times. The explained variance (R-squared) was used as test-statistics for the null distribution, and the p-value was calculated as the number of times the explained variance of the random gene-set exceeded the variance explained by 16p11.2 or 22q11.2 genes. A similar approach was performed for the individual gene correlations using median correlation as test-statistics.”

- We also added this sentence in the introduction:

“Robust functional brain networks measured by rsfMRI are also recapitulated by spatial patterns of gene expression in the adult brain ^{40,41}.”

- We added these paragraphs in the discussion:

“The spatial expression patterns of genes encompassed in both genomic loci were associated with FC-signatures of the corresponding deletion but many genes outside these 2 loci also show similar levels of association. This redundancy may represent a factor underlying shared FC signatures between both deletions.”

“Recent work has shown that the brain transcriptome follows broad spatial trends that are closely related to the functional network architecture ^{42,43} and many genes share similar spatial patterns of expression ⁴⁰. In line with these broad spatial trends, we show that FC-signatures of deletions are associated with spatial expression patterns of genes within as well as outside the genomic loci of interest. The FC profile of the thalamus, dorsal anterior cingulate, and left inferior parietal lobule were associated with expression patterns of genes at both loci and may explain in part, the FC similarities between both deletions.”

- We changed the last sentence of the abstract:

“The FC-signatures of both deletions were associated with the spatial expression pattern of genes within as well as genes outside these 2 loci. This genetic redundancy may represent a factor underlying shared FC signatures between both deletions and idiopathic conditions.”

- We added this section in the Supplementary method:

“Aligning the gene expression maps from AHBA to the MIST64 functional parcellation:

To investigate the transcriptomic relationship of altered FC in each deletion, we aligned publicly-available atlas of gene expression in the adult human cortex from the Allen Human Brain Atlas (AHBA) dataset ⁴¹ to the MIST64 brain parcellation following previously published guidelines for probe-to-gene mappings and intensity-based filtering ³⁴ and adapting the abagen toolbox ³⁵. We normalized expression values within each brain sample across genes for each of the 6 donors and then for each gene across samples for each donor using a

scaled robust sigmoid normalization. We computed the mean of the normalized values of all samples encompassed within each functional region of the MIST64. This was performed for each donor and then averaged across donors. A leave-one-donor out sensitivity analysis generated 6 expression maps. The principal components of these 6 expression maps were highly correlated (average Pearson correlation of 0.993). The same high correlation was observed for the differential Stability score (average Pearson correlation of 0.987).

Normalized gene expression value was available for each of the 15663 genes and for each of the 64 functional brain regions.”

- We edited the Supplementary material section “Objectives and methods overview”
- We changed the Supplementary table 4
- We added in the limitation section:

“Expression data were derived from 6 adult brains of the AHBA and results should be interpreted with caution.”

Question 15

Moreover, the claims made using these genes lack either novelty or proper control. For example, the authors found that genes encompassed in 16p11.2 and 22q11.2 loci are highly correlated across anatomical regions. While $R=0.67$ is a decent correlation value, is it better than a randomly selected background of 57 genes expressed in the brain? If not, what is interesting about this finding?

Response:

The question raised by the reviewer is important and we agree that this correlation is not specific to 16p11.2 and 22q11.2. To address this comment, we sampled randomly gene-sets outside the 16p11.2 and 22q11.2 locus and performed the same correlation 1000 times. It shows that the correlation is not specific to 22q11.2 and 16p11.2. The distribution of correlation over 1000 randomly sampled gene sets are displayed in Figure B below. This non-specific correlation highlights the fact that many large CNVs sampled genome-wide (random gene sets) will include genes covering the most common co-expression modules observed in the brain⁴¹. This could explain why so many genomic variants are associated with risk for the same psychiatric conditions. This figure and these results are not included in the new manuscript since responding to comment #14 of the same reviewer required completely reshaping the result section on gene expression. The low specificity is addressed in this new result section.

Figure B: Caption: Figure on the unspecific correlations between co-expression matrices of the 16p11.2 and 22q11.2 genes. The vertical red line represents the correlation between co-expression matrices of the 16p11.2 genes (n=26) and 22q11.2 genes (n=45) using AHBA data from French-Paus mapped on the Desikan atlas.⁽⁴⁴⁾. The histogram represents the null distribution, which is generated by randomly sampling 26 and 45 genes from a total of 20736 genes, and re-computing co-expression matrices and correlation between upper triangular matrices 10000 times. Y-axis: count; X-axis correlation (r) values. Vertical lines show the empirical p-value and correlation between 16p11.2 and 22q11.2 co-expression matrices.

Question 16

The authors also perform PCA on the expression levels of these genes and report that genes in PCI have higher differential stability score. While this arguably provides more interesting genes, the initial gene number of 57 is already a small set.

Response :

The reviewer is correct. The set of genes within the 22q11.2 and the 16p11.2 loci are too small to provide a clear interpretation of the relationship between DS and any particular component. This analysis is not essential to the interpretation of the gene expression results of the updated manuscript according to comment #14 of the reviewers.

Minor issues:

Question 17

Can the authors explain how the permutation p-values were calculated? The reference points to a method paper but a brief description would be better.

Response:

We edited the method as follow:

“We defined the global FC shift as the average of the β values across all 2,080 connections and tested for significance using a permutation test. We performed 5000 random CWAS by contrasting CNV carriers and controls after shuffling the genetic status labels. For example, we randomly permuted the clinical status of 16p11.2 deletion carriers and their respective controls in the 16p11.2 deletion *vs* control CWAS. We then estimated the p-value by calculating the frequency of random global FC shifts that were greater than the original observation⁴⁵. ”

Question 18

Can the authors describe which measurement FC calculations rely on? I could not find this in the supplementary method.

Response:

We added a few lines in the introduction on the definition of FC for non-experts.

“FC represents the intrinsic low-frequency synchronization between different neuroanatomical regions. It is measured by means of resting-state functional magnetic resonance imaging (rs-fMRI) which captures fluctuations of blood oxygenation as an indirect measure of neural activity across brain areas when no explicit task is performed ^{46,47}. Robust functional brain networks measured by rs-fMRI are also recapitulated by spatial patterns of gene expression in the adult brain ^{40,41}.”

Software / Code

We noticed that your paper appears to rely on the use of custom code/software. We would like to clarify if and how the software/algorithms necessary to reproduce the results will be made available to the scientific community upon publication as required by our material sharing requirements. For more information on this please see <http://www.nature.com/authors/policies/availability.html#code>

Response

We added in the paper:

“Data availability

Beta maps from all Connectome wide association studies (16p11.2 deletion and duplication, 22q11.2 deletion and duplication, ASD, SZ, and ADHD) performed in this study are available in the supplemental table 1 and on GitHub. Expression data used in this study are also available in supplemental table 4.

Code availability

The processing scripts and custom analysis software used in this work are available in a publicly accessible GitHub repository with instructions on how to set up a similar computation environment and with examples of key visualizations in the paper:

https://github.com/surchs/Neuropsychiatric_CNV_code_supplement”

References

1. Di Martino, A. *et al.* The autism brain imaging data exchange: towards a large-scale evaluation of the intrinsic brain architecture in autism. *Mol. Psychiatry* **19**, 659–667 (2014).
2. King, J. B. *et al.* Generalizability and reproducibility of functional connectivity in autism. *Mol. Autism* **10**, 27 (2019).
3. Cerliani, L. *et al.* Increased Functional Connectivity Between Subcortical and Cortical Resting-State Networks in Autism Spectrum Disorder. *JAMA Psychiatry* **72**, 767–777 (2015).
4. Tomasi, D. & Volkow, N. D. Reduced Local and Increased Long-Range Functional Connectivity of the Thalamus in Autism Spectrum Disorder. *Cereb. Cortex* **29**, 573–585 (2019).
5. Holiga, Š. *et al.* Patients with autism spectrum disorders display reproducible functional connectivity alterations. *Sci. Transl. Med.* **11**, (2019).
6. Fornito, A. & Bullmore, E. T. Reconciling abnormalities of brain network structure and function in schizophrenia. *Curr. Opin. Neurobiol.* **30**, 44–50 (2015).
7. Giraldo-Chica, M. & Woodward, N. D. Review of thalamocortical resting-state fMRI studies in schizophrenia. *Schizophr. Res.* **180**, 58–63 (2017).
8. Ferri, J. *et al.* Resting-state thalamic dysconnectivity in schizophrenia and relationships with symptoms. *Psychol. Med.* **48**, 2492–2499 (2018).
9. Padula, M. C. *et al.* Structural and functional connectivity in the default mode network in 22q11.2 deletion syndrome. *J. Neurodev. Disord.* **7**, 23 (2015).
10. Schreiner, M. J. *et al.* Default mode network connectivity and reciprocal social behavior in 22q11.2 deletion syndrome. *Soc. Cogn. Affect. Neurosci.* **9**, 1261–1267 (2014).
11. Schleifer, C. *et al.* Dissociable Disruptions in Thalamic and Hippocampal Resting-State Functional Connectivity in Youth with 22q11.2 Deletions. *J. Neurosci.* (2018)
doi:10.1523/JNEUROSCI.3470-17.2018.

12. Bertero, A. *et al.* Autism-associated 16p11.2 microdeletion impairs prefrontal functional connectivity in mouse and human. *Brain* (2018) doi:10.1093/brain/awy111.
13. Sanders, S. J. *et al.* Insights into Autism Spectrum Disorder Genomic Architecture and Biology from 71 Risk Loci. *Neuron* **87**, 1215–1233 (2015).
14. Marshall, C. R. *et al.* Contribution of copy number variants to schizophrenia from a genome-wide study of 41,321 subjects. *Nat. Genet.* **49**, 27–35 (2017).
15. Martin-Brevet, S. *et al.* Quantifying the Effects of 16p11.2 Copy Number Variants on Brain Structure: A Multisite Genetic-First Study. *Biol. Psychiatry* (2018) doi:10.1016/j.biopsych.2018.02.1176.
16. Sun, D. *et al.* Large-scale mapping of cortical alterations in 22q11.2 deletion syndrome: Convergence with idiopathic psychosis and effects of deletion size. *Mol. Psychiatry* (2018) doi:10.1038/s41380-018-0078-5.
17. *Diagnostic and Statistical Manual of Mental Disorders: Dsm-5.* (Amer Psychiatric Pub Incorporated, 2013). doi:10.1176/appi.books.9780890425596.
18. Lim, A., Hoek, H. W., Deen, M. L., Blom, J. D. & GROUP Investigators. Prevalence and classification of hallucinations in multiple sensory modalities in schizophrenia spectrum disorders. *Schizophr. Res.* **176**, 493–499 (2016).
19. Waters, F. *et al.* Visual hallucinations in the psychosis spectrum and comparative information from neurodegenerative disorders and eye disease. *Schizophr. Bull.* **40 Suppl 4**, S233–45 (2014).
20. Simon, D. M. & Wallace, M. T. Dysfunction of sensory oscillations in Autism Spectrum Disorder. *Neurosci. Biobehav. Rev.* **68**, 848–861 (2016).
21. Hippolyte, L. *et al.* The Number of Genomic Copies at the 16p11.2 Locus Modulates Language, Verbal Memory, and Inhibition. *Biol. Psychiatry* **80**, 129–139 (2016).
22. Biria, M. *et al.* Visual processing deficits in 22q11.2 Deletion Syndrome. *Neuroimage Clin* **17**, 976–986 (2018).
23. Rihs, T. A. *et al.* Altered auditory processing in frontal and left temporal cortex in 22q11.2 deletion syndrome: a group at high genetic risk for schizophrenia. *Psychiatry Res.* **212**, 141–149 (2013).

24. Cantonas, L.-M. *et al.* Abnormal development of early auditory processing in 22q11.2 Deletion Syndrome. *Transl. Psychiatry* **9**, 138 (2019).
25. Kebets, V. *et al.* Somatosensory-Motor Dysconnectivity Spans Multiple Transdiagnostic Dimensions of Psychopathology. *Biol. Psychiatry* (2019) doi:10.1016/j.biopsych.2019.06.013.
26. Gong, W. *et al.* A powerful and efficient multivariate approach for voxel-level connectome-wide association studies. *Neuroimage* **188**, 628–641 (2019).
27. Bellec, P. *et al.* Impact of the resolution of brain parcels on connectome-wide association studies in fMRI. *Neuroimage* (2015) doi:10.1016/j.neuroimage.2015.07.071.
28. Bellec, P., Rosa-Neto, P., Lyttelton, O. C., Benali, H. & Evans, A. C. Multi-level bootstrap analysis of stable clusters in resting-state fMRI. *Neuroimage* **51**, 1126–1139 (2010).
29. Urchs, S. *et al.* MIST: A multi-resolution parcellation of functional brain networks. *MNI Open Research* **1**, 3 (2019).
30. Benjamini, Y. & Hochberg, Y. Controlling the False Discovery Rate: A Practical and Powerful Approach to Multiple Testing. *J. R. Stat. Soc. Series B Stat. Methodol.* **57**, 289–300 (1995).
31. Vul, E., Harris, C., Winkielman, P. & Pashler, H. Puzzlingly High Correlations in fMRI Studies of Emotion, Personality, and Social Cognition. *Perspect. Psychol. Sci.* **4**, 274–290 (2009).
32. Chawner, S. J. R. A. *et al.* Genotype-phenotype associations in children with copy number variants associated with high neuropsychiatric risk in the UK (IMAGINE-ID): a case-control cohort study. *Lancet Psychiatry* **6**, 493–505 (2019).
33. Huguet, G. *et al.* Measuring and Estimating the Effect Sizes of Copy Number Variants on General Intelligence in Community-Based Samples. *JAMA Psychiatry* (2018) doi:10.1001/jamapsychiatry.2018.0039.
34. Arnatkevic Iūtė, A., Fulcher, B. D. & Fornito, A. A practical guide to linking brain-wide gene expression and neuroimaging data. *Neuroimage* **189**, 353–367 (2019).
35. Markello, R., Shafiei, G., Zheng, Y.-Q. & Mišić, B. *abagen: A toolbox for the Allen Brain Atlas genetics*

data. (2020). doi:10.5281/zenodo.3688800.

36. Romero-Garcia, R. *et al.* Schizotypy-Related Magnetization of Cortex in Healthy Adolescence Is Colocated With Expression of Schizophrenia-Related Genes. *Biol. Psychiatry* (2019) doi:10.1016/j.biopsych.2019.12.005.
37. Morgan, S. E. *et al.* Cortical patterning of abnormal morphometric similarity in psychosis is associated with brain expression of schizophrenia-related genes. *Proc. Natl. Acad. Sci. U. S. A.* **116**, 9604–9609 (2019).
38. Ball, G., Seidlitz, J., Beare, R. & Seal, M. L. Cortical remodelling in childhood is associated with genes enriched for neurodevelopmental disorders. *Neuroimage* **215**, 116803 (2020).
39. Seidlitz, J. *et al.* Transcriptomic and Cellular Decoding of Regional Brain Vulnerability to Neurodevelopmental Disorders. doi:10.1101/573279.
40. Richiardi, J. *et al.* BRAIN NETWORKS. Correlated gene expression supports synchronous activity in brain networks. *Science* **348**, 1241–1244 (2015).
41. Hawrylycz, M. *et al.* Canonical genetic signatures of the adult human brain. *Nat. Neurosci.* **18**, 1832 (2015).
42. Fornito, A., Arnatkevičiūtė, A. & Fulcher, B. D. Bridging the Gap between Connectome and Transcriptome. *Trends Cogn. Sci.* **23**, 34–50 (2019).
43. Wei, Y. *et al.* Genetic mapping and evolutionary analysis of human-expanded cognitive networks. *Nat. Commun.* **10**, 4839 (2019).
44. French, L. & Paus, T. A FreeSurfer view of the cortical transcriptome generated from the Allen Human Brain Atlas. *Front. Neurosci.* **9**, 323 (2015).
45. Phipson, B. & Smyth, G. K. Permutation P-values should never be zero: calculating exact P-values when permutations are randomly drawn. *Stat. Appl. Genet. Mol. Biol.* **9**, Article39 (2010).
46. Biswal, B., Yetkin, F. Z., Haughton, V. M. & Hyde, J. S. Functional connectivity in the motor cortex of resting human brain using echo-planar MRI. *Magn. Reson. Med.* **34**, 537–541 (1995).
47. van den Heuvel, M. P. & Hulshoff Pol, H. E. Exploring the brain network: a review on resting-state fMRI functional connectivity. *Eur. Neuropsychopharmacol.* **20**, 519–534 (2010).

REVIEWER COMMENTS

Reviewer #1 (Remarks to the Author):

The authors have comprehensively addressed my comments in the revised manuscript and response letter. I have no further concerns.

Clare Kelly

Reviewer #2 (Remarks to the Author):

The authors have fully addressed all of my review comments. I do not have additional comments.

Reviewer #3 (Remarks to the Author):

The authors have been very responsive to my previous concerns and I appreciate the changes made. I have just a few minor questions/comments that would be great to have addressed by text additions/changes:

1- If I understood correctly, predictor variables of PLSR analysis were gene expression values and the response variable was the FC between the region that is being tested and other regions (one region per row). A significant p-value was reported for the tested region if the genes predicted that region's FC pattern better than randomized FC signatures.

Interestingly, many significant regions are similar when the genes in 16p11.2 and 22q11.2 are switched to predict FC pattern in 16p11.2 and 22q11.2. But do the authors know if this is different than randomized gene selection? Perhaps randomized selection would also highlight the same regions? It appears the methods section already has this analysis for PLSR:

"To test the specificity of the relationship between gene expression and FC, we randomly sampled 10000 gene sets (n=24 for 16p11.2 genes and n=37 for 22q11.2 genes) from 15633 genes and re-computed the PLSR 10000 times. The explained variance (R-squared) was used as test-statistics for the null distribution, and the p-value was calculated as the number of times the explained variance of the random gene-set exceeded the variance explained by 16p11.2 or 22q11.2 genes."

However, I did not see the results of such analysis reported in the results section of the manuscript. Given that the p-value for PLSR was calculated by randomizing FC values and not by randomizing the gene expression values, it would be interesting to know which regions are specific to which CNV and which are shared based on randomized gene expression background.

2- While the genome wide correlation analysis has some interesting results, the methodology is not very detailed and the results are not interpreted in much detail.

For methodology: Was correlation performed for global connectivity of each region? The expression values from AHBA should be from 6 individuals, was mean expression taken across individuals? How was the expression data handled? Was the expression data z-transformed (otherwise spearman correlation would be a better option)?

For interpretation: The results show no significant genes for 16p11.2 but hundreds of genes for 22q11.2. How do the authors interpret this?

3- The authors wrote: "However, both deletions were enriched in genes..."

Enrichment typically implies statistical analysis of overlap. If that is not the case, please change the wording.

Reviewer #1 (Remarks to the Author):

The authors have comprehensively addressed my comments in the revised manuscript and response letter. I have no further concerns.

Clare Kelly

Reviewer #2 (Remarks to the Author):

The authors have fully addressed all of my review comments. I do not have additional comments.

Reviewer #3 (Remarks to the Author):

The authors have been very responsive to my previous concerns and I appreciate the changes made. I have just a few minor questions/comments that would be great to have addressed by text additions/changes:

Question 1

If I understood correctly, predictor variables of PLSR analysis were gene expression values and the response variable was the FC between the region that is being tested and other regions (one region per row). A significant p-value was reported for the tested region if the genes predicted that region's FC pattern better than randomized FC signatures.

Interestingly, many significant regions are similar when the genes in 16p11.2 and 22q11.2 are switched to predict FC pattern in 16p11.2 and 22q11.2.

But do the authors know if this is different than randomized gene selection? Perhaps randomized selection would also highlight the same regions? It appears the methods section already has this analysis for PLSR: "To test the specificity of the relationship between gene expression and FC, we randomly sampled 10000 gene sets (n=24 for 16p11.2 genes and n=37 for 22q11.2 genes) from 15633

genes and re-computed the PLSR 10000 times. The explained variance (R-squared) was used as test-statistics for the null distribution, and the p-value was calculated as the number of times the explained variance of the random gene-set exceeded the variance explained by 16p11.2 or 22q11.2 genes.” However, I did not see the results of such analysis reported in the results section of the manuscript. Given that the p-value for PLSR was calculated by randomizing FC values and not by randomizing the gene expression values, it would be interesting to know which regions are specific to which CNV and which are shared based on randomized gene expression background.

Response:

Yes, the reviewer is correct, the predictor variables of PLSR analysis were gene expression values, and the response variable was the FC between the region that is being tested and other regions (one region per row).

Similar to the results reported for correlations with individual genes, we find that genes outside both CNV loci (Genome-wide randomly sampled gene sets) can explain similar percentages of variance in PLSR model for both global and regional FC signatures.

We added the following sentence in the result section::

“Overall, this relationship was repeatedly observed across the genome: a large proportion of gene sets (randomly sampled genome-wide, n=37 and 24) showed a similar level of association with either global or regional FC-signature of both deletions especially for the 22q11.2.”

In addition, we have included the code and results for assessing the specificity using RandomGeneSets in the GitHub repository (see function: script1_call_PLSR_nodal_and_regional.m).

Question 2

While the genome-wide correlation analysis has some interesting results, the methodology is not very detailed and the results are not interpreted in much detail.

2.1 Was correlation performed for global connectivity of each region?

Response:

The correlation was indeed performed between gene expression and the mean connectivity of each region. This information is available in the “*Gene expression analyses method*”:

“Expression values were associated with the average connectivity alteration of the corresponding regions (specifically, we computed the mean of all 64 beta values of each region, a measure of global connectivity).”

2.2 The expression values from AHBA should be from 6 individuals, was mean expression taken across individuals? How was the expression data handled? Was the expression data z-transformed (otherwise spearman correlation would be a better option)?

Response:

The reviewer is correct. We computed the mean of the normalized values of all samples encompassed within each functional region of the MIST64 functional atlas. This was performed for each donor and then averaged across donors according to previously published methods ¹⁶.

We moved the section previously in the supplemental methods to the methods of the main text to provide the details requested by the reviewer:

“...Aligning the gene expression maps from AHBA to the MIST64 functional parcellation

To investigate the transcriptomic relationship of altered FC in each deletion, we aligned gene expression values in the adult human brain from the Allen Human Brain Atlas (AHBA) dataset ¹⁵ to the MIST64 brain parcellation following previously published guidelines for probe-to-gene mappings and intensity-based filtering ¹⁶ and adapting the abagen toolbox ¹⁷. We normalized expression values

within each brain sample across genes for each of the 6 donors and then for each gene across samples for each donor using a scaled robust sigmoid normalization ¹⁶. We computed the mean of the normalized values of all samples encompassed within each functional region of the MIST64. This was performed for each donor and then averaged across donors. A leave-one-donor out sensitivity analysis generated 6 expression maps. The principal components of these 6 expression maps were highly correlated (average Pearson correlation of 0.993). The same high correlation was observed for the differential Stability score (average Pearson correlation of 0.987).

Normalized gene expression value was available for each of the 15663 genes and for each of the 64 functional brain regions...”

2.3 For interpretation: The results show no significant genes for 16p11.2 but hundreds of genes for 22q11.2. How do the authors interpret this?

Response

This observation is indeed intriguing and important.

We added the following interpretation in the discussion :

“...Expression data were derived from 6 adult brains of the AHBA. Gene expression studies reported a later window of susceptibility in SZ compared to ASD ⁵⁵. The 22q11.2 deletion may, therefore, be in part driven by later onset mechanisms, which could explain why its FC-signature correlates to the adult spatial expression patterns of so many genes. On the other hand, the 16p11.2 deletion which is preferentially associated with autism may be linked to earlier developmental mechanisms which may explain why its FC-signature is much less correlated to patterns of adult brain expression. further analyses are required to understand the potential relationship between regional connectivity and temporal gene expression....”

Question 3

The authors wrote: “However, both deletions were enriched in genes...”

Enrichment typically implies statistical analysis of overlap. If that is not the case, please change the wording.

Response:

We agree with the reviewer. We modified the text as followed:

“However, genes with correlations ranking higher than the genome-wide 98th percentile were over-represented in both deletions”

CONSORTIA

We note that your study has one or more consortia included in the author list. In order for a consortium to appear in the main author list, at least one member of the consortium must be considered as an author on the manuscript. All members of the consortium who are considered authors must be listed in the published article. If not all members of the consortium agree to the responsibilities of authorship, the remaining, non-author members of the consortium can be listed separately in the Supplementary Note. If no individual member of a consortium can be considered an author on the manuscript, the consortium should be listed in the Acknowledgments section.

Response:

We deleted the *Simons Variation in Individuals Project Consortium* from the author list and we added the following information in the acknowledgement section:

“...We are grateful to all of the families at the participating Simons Simplex Collection (SSC) sites, as well as the principal investigators (A. Beaudet, R. Bernier, J. Constantino, E. Cook, E. Fombonne, D. Geschwind, R. Goin-Kochel, E. Hanson, D. Grice, A. Klin, D. Ledbetter, C. Lord, C. Martin, D.

Martin, R. Maxim, J. Miles, O. Ousley, K. Pelphey, B. Peterson, J. Piggot, C. Saulnier, M. State, W. Stone, J. Sutcliffe, C. Walsh, Z. Warren, E. Wijsman). We appreciate obtaining access to imaging and phenotypic data on SFARI Base. Approved researchers can obtain the Simons Variation in Individuals Project population dataset described in this study by applying at <https://base.sfari.org...>”

REVIEWERS' COMMENTS

Reviewer #3 (Remarks to the Author):

Thank you for the additional responses, all of my questions have been answered.